



# UKESM1.1: Development and evaluation of an updated configuration of the UK Earth System Model

Jane P. Mulcahy[1], Colin G. Jones[2,3], Steven T. Rumbold[2,4], Till Kuhlbrodt[2,4], Andrea J. Dittus[2,4], Edward W. Blockley[1], Andrew Yool[5], Jeremy Walton[1], Catherine Hardacre[1], Timothy Andrews[1], Alejandro Bodas-Salcedo[1], Marc Stringer[2,4], Lee de Mora[6], Phil Harris[7], Richard Hill[1], Doug Kelley[7], Eddy Robertson[1], and Yongming Tang[1]

[1]Met Office Hadley Centre, Exeter, UK
[2]National Centre for Atmospheric Science, UK
[3]School of Earth and Environment, University of Leeds, UK
[4]Department of Meteorology, University of Reading, Reading, UK
[5]National Oceanography Centre, Southampton, UK
[6]Plymouth Marine Laboratory, Plymouth, UK
[7]UK Centre for Ecology and Hydrology, Wallingford, UK

**Correspondence:** Jane P. Mulcahy (jane.mulcahy@metoffice.gov.uk)

**Abstract.** Many CMIP6 models exhibit a substantial cold bias in global mean surface temperature (GMST) in the latter part of the 20th century. An overly strong negative aerosol forcing has been suggested as a leading contributor to this bias. An updated configuration of UKESM1, UKESM1.1, has been developed with the aim of reducing the historical cold bias in this model. Changes implemented include an improved representation of $SO_2$ dry deposition along with several other smaller

modifications to the aerosol scheme and a retuning of some uncertain parameters of the fully coupled Earth System Model. The Diagnostic, Evaluation and Characterization of Klima (DECK) experiments, a 6-member historical ensemble and a subset of future scenario simulations are completed. In addition, the total anthropogenic effective radiative forcing (ERF), its components and the effective and transient climate sensitivities are also computed. The UKESM1.1 pre-industrial climate is warmer than UKESM1 by up to 0.75 K and a significant improvement in the historical GMST record is simulated with the magnitude of the

cold bias reduced by over 50%. The warmer climate increases ocean heat uptake in the northern hemisphere oceans and reduces Arctic sea ice in better agreement with observations. Changes to the aerosol and related cloud properties are the key drivers of the improved GMST simulation despite only a modest change in aerosol ERF (+0.08 Wm$^{-2}$). The total anthropogenic ERF increases from 1.76 Wm$^{-2}$ in UKESM1 to 1.84 Wm$^{-2}$ in UKESM1.1. The effective climate sensitivity (5.27 K) and transient climate response (2.64 K) remain largely unchanged from UKESM1 (5.36 K and 2.76 K respectively).



# 1  Introduction

The ability of a global climate model (GCM) to accurately simulate the historical climate is generally regarded as an important indicator of the model's potential skill in simulating future climate. In particular, the historical global mean surface temperature is a widely used metric in assessing the performance of global climate models despite this not necessarily directly translating to skillful future climate projections (Kiehl, 2007). A number of modelling centres employ various tuning practices to achieve good agreement with the observed historical temperature record during the model development cycle (Mauritsen and Roeckner,

2020; Hourdin et al., 2017; Schmidt et al., 2017) while others treat it as an emergent model property (Senior et al., 2020). Notwithstanding such practices many models participating in the 6th Coupled Model Intercomparison Project (CMIP6) have a significant cold bias in the historical global mean surface temperature through the latter half of the 20th century (Flynn and Mauritsen, 2020). An overly strong aerosol forcing has been suggested as a leading candidate responsible for this bias consistent with increasing anthropogenic aerosol emissions during this period (Flynn and Mauritsen, 2020; Dittus et al., 2020;

Andrews et al., 2020). In particular, anthropogenic emissions of sulphur dioxide ($SO_2$), a precursor to sulphate aerosol, was the predominant source of anthropogenic aerosols during this time. Sulphate aerosol directly scatters incoming shortwave solar radiation and can also influence cloud albedo and lifetime by acting as efficient cloud condensation nuclei. After 1980 with the widespread implementation of clean air legislation $SO_2$ emissions steadily declined and the climate has subsequently warmed in response to greenhouse gas emissions.

UKESM1 has a particularly large cold bias peaking in the 1970-1980 period with a negative bias approaching 0.5K (Sellar et al., 2019). Investigations by Hardacre et al. (2021) highlight a significant positive bias in the surface concentration of $SO_2$ in UKESM1 at measurement sites across the historically large emission source regions of Europe and northeastern USA. Interestingly a low bias in surface $SO_4$ concentrations is found in the same regions (Mulcahy et al., 2020; Hardacre et al., 2021). More generally, Earth System models (ESMs) with more complex chemistry and aerosol schemes have a larger cold bias in

CMIP6 than their physical model counterparts (Zhang et al., 2021). A strong correlation is found between the anthropogenic sulphate burden and surface temperature change over the historical period, further strengthening the argument for the role of an overly strong sulphate aerosol forcing in these models (Zhang et al., 2021).

Given the likely strong contribution of aerosol to the historical surface temperature bias a number of key questions arise regarding the simulation of $SO_2$, the pathways leading to sulphate aerosol and associated aerosol-climate interactions in

UKESM1. Specifically, are the emissions of anthropogenic $SO_2$ too high in the CMIP6 forcing dataset (McDuffie et al., 2020; Aas et al., 2019) or are the sinks of $SO_2$ realistically simulated in the model? With respect to the latter we refer to wet and dry deposition as well as loss via the chemical oxidation of $SO_2$ to sulphate aerosol. If sink processes are too low the atmospheric residence time of $SO_2$ will be too long, resulting in excess $SO_2$ being transported away from industrial source regions and being oxidised to form aerosol in more pristine remote regions which are more susceptible to aerosol forcing (Carslaw et al.,

2013). Evidence suggests that a large fraction (up to 50%) of emitted anthropogenic $SO_2$ is deposited within 200 to 300 km of emission sources with 30-35% dry deposited and 10-15% oxidised to $SO_4$ (Smith and Jeffrey, 1975; Wys et al., 1978).





In coarse resolution climate models like UKESM1 this is the equivalent to two or three model grid-boxes and so accurately representing these sub-grid processes is challenging in GCMs.

Following the above hypothesis, Hardacre et al. (2021) examine the impact of an updated parameterization for the dry deposition of $SO_2$ on the surface $SO_2$ concentration bias in UKESM1. The new parameterization takes into account whether the surface vegetation is wet or dry when calculating the surface resistance to species uptake. Due to the high solubility of $SO_2$, the wetter and more humid at the surface the higher the uptake of $SO_2$. The new parameterization leads to a significant improvement (of the order of 50%) in the $SO_2$ bias against ground-based observations in the above study. While reductions of $SO_2$ close to source further degrade the pre-existing low bias in $SO_4$ (Mulcahy et al., 2020) it is possible that model process deficiencies in the conversion of $SO_2$ to $SO_4$ also exist. Mulcahy et al. (2020) also noted that $SO_4$ production in regions such as Europe appeared to be more oxidant limited than $SO_2$ limited. Interestingly, Hardacre et al. (2021) show a larger reduction in surface $SO_2$ and $SO_4$ remote from source (e.g. over the North Atlantic region) than over the source regions supporting the hypothesis that excess $SO_2$ drives remote aerosol loading and subsequent forcing.

In this paper we document and characterize a new science configuration of the UKESM model which we refer to as UKESM1.1. While the revised $SO_2$ dry deposition parameterization comprises the main science development in UKESM1.1 we also incorporate a number of smaller science changes resulting from the extensive evaluation of the UKESM1 model (e.g. Sellar et al. (2019); Mulcahy et al. (2020); Yool et al. (2019); Robson et al. (2020)). In addition, we revise some of the specific tunings applied to the coupled aspects of UKESM as outlined in Sellar et al. (2019). In Section 2 we describe the UKESM1.1 science configuration, detailing all model changes and motivations for such changes. The impact of the new configuration on a number of key climate-based metrics are assessed in Section 4. We focus our evaluation on the fully coupled historical simulations and compare the present-day UKESM1.1 climate to its predecessor, UKESM1 and a wide range of observations where available. We then assess of the impact of the new model on the anthropogenic forcing, the transient and effective climate sensitivities. Finally, we compare the future climate response to UKESM1.

## 2 Model description

The UKESM1 model is described in detail in Sellar et al. (2019) and so we provide only an overview of its components here. UKESM1 comprises the global coupled atmosphere-ocean climate model, HadGEM3-GC3.1 (Williams et al., 2017) along with additional Earth system components which are important because of their feedbacks on the climate system. These include the simulation of the uptake of carbon and nitrogen in marine and terrestrial ecosystems and the interactive representation of trace gas and aerosol composition changes as well as their interactions within the full Earth system.

The atmosphere component of UKESM1 uses the Global Atmosphere 7.1 (GA7.1) science configuration of the Unified Model (Walters et al., 2019; Mulcahy et al., 2018) and has a horizontal resolution of approximately 135 km (1.25 °x1.875 °) and 85 vertical levels with a terrain-following hybrid-height coordinate, reaching up to 85 km. Aerosols and trace gas chemistry are simulated in UKESM1 using the stratospheric-tropospheric configuration of the United Kingdom Chemistry and Aerosol





(UKCA) model (Archibald et al., 2020) which is coupled to the GLOMAP-mode 2-moment modal aerosol scheme (Mulcahy
et al., 2020, 2018). The aerosol model component and changes implemented in this work are described in more detail below.

In the ocean UKESM1 uses the low resolution version ($1\,^{\circ}$, Kuhlbrodt et al. (2018)) of the Nucleus for European Modelling
of the Ocean (NEMO) (Storkey et al., 2018) with 75 vertical levels. Sea ice is simulated using the Los Alamos Sea Ice Model
(CICE, Ridley et al. (2018)). A small correction to the coupled sea-ice heat fluxes in NEMO is applied here. This corrects an
inconsistent use of sea ice fraction in the calculation of sea ice grid box mean values (used in the CICE component) from sea
ice area mean values (used in the atmosphere component).

The land surface component uses the Joint UK Land Environment Simulator (JULES) model (Best et al., 2011; Clark et al.,
2011) and shares the same latitude-longitude grid as the atmosphere. Surface and sub-surface runoff is transported to the ocean
using the TRIP river routing scheme (Total Runoff Integrating Pathways, Oki and Sud (1998)). Terrestrial biogeochemistry
is simulated in UKESM1 by coupling JULES to the dynamic vegetation model, TRIFFID (Cox, 2001) and the RothC soil
carbon model (Coleman and Jenkinson, 1999). Updates made to these schemes for inclusion into UKESM1 are described in
Sellar et al. (2019). Noteworthy developments included the extension of plant functional types to better distinguish between
evergreen and deciduous plants and between tropical and temperate evergreen plants, the inclusion of a nitrogen scheme which
limits terrestrial carbon uptake and extensions to the representation of land use change.

Marine biogeochemistry is simulated using the Model of Ecosystem Dynamics nutrient Utilisation, Sequestration and Acid-
ification (MEDUSA, Yool et al. (2013, 2021)). MEDUSA is a medium complexity plankton ecosystem model representing the
biogeochemical cycles of nitrogen, silicon and iron nutrients, as well as carbon, alkalinity and dissolved oxygen.

## 2.1 Aerosol developments

The GLOMAP-mode aerosol scheme represents the emissions, atmospheric evolution and deposition of sea salt, sulphate
($SO_4$), black carbon and organic carbon species. Full details of the aerosol scheme as implemented in UKESM1 and a detailed
evaluation of the historical aerosol simulation, as run for CMIP6, are given in Mulcahy et al. (2020).

A number of developments to the $SO_2$ dry deposition scheme are implemented. These are described in detail in Hardacre
et al. (2021) so we only briefly outline them here. The dry deposition scheme in UKCA follows that of Wesely (1989). This
uses a resistance-based approach to calculate the dry deposition velocity, $v_d$, as:

$$v_d = \frac{1}{r_a + r_b + r_c} \tag{1}$$

where $r_a$ represents the aerodynamic resistance, $r_b$ is the quasi-laminar sublayer resistance, and $r_c$ represents the surface
resistance. The dry deposition developments in this work apply to the surface resistance term, $r_c$, which is sensitive to the
individual surface properties and surrounding climatic conditions. For non-vegetated surfaces (e.g. open water, bare soil or
snow covered surfaces), $r_c$ is set at a suitable global constant. Over vegetated surfaces $r_c$ is calculated as a function of the
stomatal resistance ($R_{stom}$), the canopy cuticle resistance ($R_{cut}$), and the soil resistance ($R_{soil}$). $R_{soil}$ and $R_{cut}$ are defined
for the 13 fractional land cover types in the model. In UKESM1 the calculation of $R_{cut}$ and $R_{soil}$ has no dependence on
whether the underlying vegetation is wet or dry nor do they depend on the near surface climate conditions. Given the high





**Table 1.** Summary of the DMS chemical reactions in UKESM1 and UKESM1.1

| UKESM1 | UKESM1.1 |
|---|---|
| $DMS + OH \rightarrow SO_2$ | $DMS + OH \rightarrow SO_2$ |
| $DMS + OH \rightarrow SO_2 + MSA$ | $DMS + OH \rightarrow 0.6SO_2 + 0.4DMSO$ |
| $DMS + NO_3 \rightarrow SO_2$ | $DMS + NO_3 \rightarrow SO_2$ |
| $DMS + O(^3P) \rightarrow SO_2$ | $DMSO + OH \rightarrow 0.6SO_2$ |
| | $DMS + O(^3P) \rightarrow SO_2$ |

solubility of $SO_2$ this is expected to lead to $r_c$ values which are too high and subsequently the dry deposition of this gas will be underestimated (Hardacre et al., 2021). In the updated scheme, the modelled precipitation is used to determine the degree of surface wetness. $R_{cut}$ and $R_{soil}$ for $SO_2$ are then parameterized as a function of the surface wetness, near surface relative humidity and temperature following Erisman and Baldocchi (1994); Erisman et al. (1994). Therefore rather than having a fixed

$r_c$ value for each vegetation type, $r_c$ now varies as a function of the near surface conditions, with $r_c$ being at its minimum when a surface grid-box is classified as wet.

    In addition, while analysing the UKESM1 $SO_2$ dry deposition scheme an error in the value $r_c$ over ocean surfaces was uncovered. This value had been incorrectly set to $148.9 \, \mathrm{sm^{-1}}$ during the development of UKESM1 instead of $10 \, \mathrm{sm^{-1}}$ as set in the physical model (HadGEM3-GC3.1) and used in Mulcahy et al. (2018). Numerous studies (e.g. Garland (1978), Erisman

et al. (1994), Zhang et al. (2003)) indicate the resistance to $SO_2$ deposition over the open ocean is minimal. Therefore, in addition to introducing the extension to $SO_2$ dry deposition over land surfaces as described above, we assign a value of $1 \, \mathrm{sm^{-1}}$ to open water surface fractions, noting this is different to the value used in GC3.1.

    Finally, a number of additional bugs in the aerosol model which were uncovered after the freeze of UKESM1 are corrected. The tropospheric chemistry of dimethyl sulfide (DMS) in the full stratospheric-tropospheric version of UKCA was previously

simplified to offset some of the additional computational cost of extending the chemistry in the stratosphere (Archibald et al., 2020). This resulted in a different set of chemical reactions to that in the previously used tropospheric only scheme (O'Connor et al., 2014) and the offline oxidant scheme used in GC3.1 (Mulcahy et al., 2020). The current DMS chemistry produces more sulphur atoms than are actually available and neglects the sulphur sink via dimethyl sulfoxide (DMSO), despite DMSO being an advected tracer. In UKESM1.1, the DMS chemistry is updated as shown in Table 1 and is now consistent with GC3.1.

A correction to the updating of sulphuric acid ($H_2SO_4$) first reported in Ranjithkumar et al. (2021) is also applied. In GLOMAP-mode the chemistry and aerosol timestep is currently one hour but micro-physical processes such as condensation, nucleation and coagulation occur on a shorter four minute sub-step. The concentration of $H_2SO_4$, required for both condensation and nucleation, should also have been updated on the shorter timestep but was only being produced on the longer timestep in UKESM1. This led to an erroneously high number concentration of small particles produced from the binary homogeneous

nucleation of sulphuric acid. This is corrected here with the primary effect being to reduce the nucleation mode of $SO_4$ aerosol in the free troposphere.





**Table 2.** Summary of tuned parameters in UKESM1 and UKESM1.1

| Model component | Parameter description | UKESM1 | UKESM1.1 |
|---|---|---|---|
| JULES | Albedo of snow on sea ice | albsnowv_cice[a]=0.96<br>albsnowi_cice[a]=0.68 | albsnowv_cice[a]=0.98<br>albsnowi_cice[a]=0.70 |
| JULES | Burial of vegetation by snow | n_lai_exposed[b]=1000 | n_lai_exposed[b]=27 |
| UM | Tunable parameters of dust scheme | horiz_d=10[c]<br>sm_corr=0.8[c]<br>us_am=1.1[c] | horiz_d=5[c]<br>sm_corr=0.7[c]<br>us_am=1.3[c] |
| UM | Gravity wave drag | USSP_launch_factor=1.3 | USSP_launch_factor=1.58 |
| UM | Fractional standard deviation of sub-grid cloud water content | two_d_fsd_factor=1.48 | two_d_fsd_factor=1.49 |

[a] Visible and near-infrared albedos for snow on sea ice; [b] n_lai_exposed was changed for grasses and temperate broadleaf evergreen trees only; [c] multiplicative scale factors for the horizontal dust emission flux, the frictional velocity and soil moisture.

A further correction was applied to the UKCA-Activate scheme (West et al., 2014) which parameterizes the activation of aerosol particles into cloud droplets. In this scheme the activated cloud droplet number concentration per $m^3$ is replicated upward from cloud base in contiguous clouds. This should rather be in units of number per $kg$ of air to correctly replicate the expansion of a rising air parcel with height. Correcting this bug reduces $N_d$ in contiguous clouds, in particular deep clouds.

## 2.2 Tuning of the coupled Earth system

We revisit some of the tuned parameters of the fully coupled system previously described in Kuhlbrodt et al. (2018) and Sellar et al. (2019). For the purposes of this tuning we make use of the atmosphere-only, UKESM1-AMIP, configuration in most cases (apart from the tuned sea ice model parameters as described in Section 2.2.1) which allows us to more easily evaluate the impact of the tuning on present-day metrics where more observations are available. As already mentioned above the tuning of the UKESM1 configuration was done in a pre-industrial climate only. A summary of the final tuned values in UKESM1 and UKESM1.1 are provided in Table 2.

### 2.2.1 Albedo of snow on sea ice

In UKESM1 the albedo of snow on sea ice was decreased by 2 percentage points (Kuhlbrodt et al., 2018) to compensate for deficient transport of warm Atlantic Ocean water into the Arctic Ocean in the lower resolution ocean model. As we will later show, the $SO_2$ changes generally warm the climate, in particular the northern hemisphere, and so we revert to the original value





as set in the 0.25 degree model (HadGEM3-GC3.1-MM, Williams et al. (2017)). The new and old values are specified in Table 2.

### 2.2.2 Burial of vegetation by snow

Another parameter that required tuning in the fully coupled ESM is the parameter, *n_lai_exposed*, which controls the degree to which vegetation is buried by snow. This was tuned in UKESM1 to correct for biases in the net surface shortwave (SW) radiation introduced when the interactive vegetation scheme was coupled to the physical model, GC3.1. When evaluated in present-day simulations however, this tuning appears to lead to an excessive burial by snow and results in a net positive bias in the snow metric used in Sellar et al. (2019) (defined as the top-of-atmosphere (TOA) outgoing clear-sky SW radiation over

land in DJF between $30\,^{\circ}$ N and $60\,^{\circ}$ N). As the climate warms with rising $CO_2$ concentrations the excessively bright snow-covered grassland is replaced by darker forests. This positive feedback was found to increase the effective climate sensitivity in UKESM1 (Andrews et al., 2019). In UKESM1.1, *n_lai_exposed* for grasses and temperate broadleaf evergreen trees has been reduced from 1000 to 27 in order to reduce the extent of the snow burial of vegetation. The retuning of *n_lai_exposed* reduces the SW snow metric from above to below the observed range (see Supplementary Table S1). We note Sellar et al.

(2019) refer to this target range as a "crude" representation of the true observational uncertainty. The UKESM1.1 performance for this metric is therefore as acceptable as UKESM1 and remains an improvement over the untuned model.

### 2.2.3 Dust

While the total aerosol optical depth (AOD) in UKESM1 compares generally well against observations, the mineral dust optical depth (DOD) is biased low (Mulcahy et al., 2020; Checa-Garcia et al., 2021). This is despite UKESM1 having larger

dust emissions and comparable dust burdens compared to other CMIP6 models (Checa-Garcia et al., 2021).

One of the key factors influencing the disparity in dust between models is the particle size distribution. UKESM1 has a significant portion of its dust in the super-coarse particle size range. Such particles have a very short atmospheric lifetime and are not optically efficient, resulting in lower DOD compared to other models which have a larger fraction of dust in the accumulation mode. As reported in Sellar et al. (2019) the mineral dust scheme (Woodward, 2001, 2011) used in UKESM1 is

highly sensitive to a number of parameters, including the vegetation distribution influencing the bare soil fraction available for dust erosion, soil moisture and wind speed. In UKESM1 all these elements are dynamic and evolve in response to the changing climatic conditions. Sellar et al. (2019) documents the three tuned parameters in UKESM1. Here we revise this dust tuning with the aim of increasing the DOD to improve agreement with ground-based and satellite AOD measurements. Table 2 specifies the original parameter values in UKESM1 and the revised values in UKESM1.1. The revised tuned parameters lead to a slight

reduction in global dust emissions of approximately 10% but the DOD is nearly doubled while the total AOD is increased by 10-20% (see Supplementary Fig. S1). These changes are consistent with a reduction in the *horiz_d* parameter which scales the total emission and a shift in the dust size distribution to the smaller sizes as a result of changes to the parameters controlling the sensitivity of soil moisture (*sm_corr*) and frictional velocity (*us_am*). AOD increases are found primarily over the Sahara, Saudi Arabia and India with much smaller increases simulated over Australia (see Supplementary Fig. S1). The spatial distribution





of the AOD is also in much better agreement with satellite retrievals of AOD from the MODIS and MISR sensors, in particular over the Sahara desert and in the dust outflow regions west of Africa. However, positive AOD biases over India are exacerbated. With more dust particles residing in the accumulation mode the potential lifetime of dust will increase with implications for the dust transport and subsequent iron deposition to the ocean in the fully coupled system. The scaling factor converting dust deposition into iron addition to the ocean was therefore also tuned so that approximately $2.6 \, \mathrm{Gmol \, Fe \, yr^{-1}}$ is added to the

ocean by this process (Yool et al., 2013). The same approach was used during the development and tuning of UKESM1 (Sellar et al., 2019).

### 2.2.4 QBO

Tuning of the parameter that controls the flux of sub-grid gravity waves generated by non-orographic sources was erroneously omitted in UKESM1. As a consequence the period of the tropical quasi-biennial oscillation (QBO) was found to be too low

in UKESM1 when compared against reanalyses (Richter et al., 2020). This has implications for the southern hemisphere climate in particular. The flux of the parameterized convectively generated wave momentum needs to be tuned depending on the model's wind and thermal structure. To do this we tune the launched gravity waves such that the QBO gives a reasonable comparison with reanalyses.

UKESM1 has a mean period of 43 months compared with 28 months in MERRA reanalysis (see Supplementary Fig. S2).

In UKESM1.1 the mean period is 28 months in agreement with the MERRA reanalysis (Fig. S2).

### 2.2.5 Radiation balance

Once the science configuration of UKESM1.1, including the model component tunings described above, was finalised a final tuning of the net radiation at the top of the atmosphere was required to ensure it is in balance. This retuning was done on a fully coupled pre-industrial control simulation and was achieved by adjusting the *two_d_fsd_factor* parameter. This parameter

scales the fractional standard deviation of the cloud water content in a grid-box as seen by the radiation scheme (Hill et al., 2015). Only a small retuning of this parameter - from a value of 1.48 to 1.49 - was required.

### 3  Model simulations

All model simulations assessed in this study follow the CMIP6 protocols as outlined in Eyring et al. (2016). While tuning of the modified parameters discussed above predominantly made use of the computationally faster atmosphere-only (AMIP)

simulations the development of UKESM1.1 was conducted in a fully coupled pre-industrial (PI) climate driven by 1850 forcing conditions. A continuously evolving PI control climate state was simulated with each change added incrementally. Once the final configuration of the model was frozen a further 462 simulated years of the PI control (*piControl*) were run. A 6-member historical ensemble was run with transient forcing from 1850 to 2014. The initial conditions (including atmosphere, ocean, sea-ice and land states) for each member of the ensemble were branched from the piControl and were equally spaced 40

years apart. In addition, the full set of Diagnostic, Evaluation and Characterization of Klima (DECK) experiments (Eyring





et al., 2016) were conducted. These comprise a pre-industrial control simulation (*piControl*) already discussed above; an *abrupt4xCO2* simulation in which the PI climate is forced by an abrupt quadrupling of $CO_2$ and run for 150 years; a *1pctCO2* experiment where a 150 year PI simulation is forced by a 1% per year increase in $CO_2$; and finally an AMIP simulation covering the period 1979 to 2014. In the AMIP simulation the sea surface temperature and sea ice conditions are prescribed from
observations (Durack and Taylor, 2017) while other atmosphere-ocean and atmosphere-land coupled variables are replaced with climatological forcing data taken from the first member of the historical ensemble. Specifically, in UKESM, the prescribed fields include transient annual mean vegetation fractions, a monthly mean climatology of the leaf area index, seawater DMS and sea surface chlorophyll concentrations and an annual mean climatology of canopy height.

An atmosphere-only version of the piControl, *piClim-control*, was also used to determine the anthropogenic effective radia-
tion forcing (ERF) as well as other key components of the anthropogenic ERF including the ERF due to anthropogenic aerosol and land use change. This configuration follows the Radiative Forcing Model Intercomparion Project (RFMIP, Pincus et al. (2016)) protocol and is essentially the AMIP configuration but driven by 1850 forcing conditions. In order to determine the total anthropogenic ERF a second piClim-control experiment was run but with all anthropogenic forcings taken from the year 2014. Anthropogenic aerosol emissions only were perturbed for the aerosol ERF perturbation simulation (*piClim-aer*). For the
ERF due to land use change (*piClim-LU*), an additional historical simulation was run which kept all forcing data apart from land use at 1850 conditions. This provides present-day vegetation distributions that resulted from land-use change only and excludes any climate-driven changes in vegetation. Following O'Connor et al. (2021) we run the ERF simulations for 45 years with the ERFs calculated from the final 30 years of the simulation.

In addition we have conducted a smaller set (3-member ensemble) of future projection simulations following the future
emission pathways as specified in ScenarioMIP (O'Neill et al., 2016). Three of the UKESM1.1 historical members were extended to 2100 following either the SSP1-2.6 or SSP3-7.0 emission pathways. SSP1-2.6 was designed to limit warming to $2\,°C$ at 2100 relative to pre-industrial values. SSP1-2.6 therefore includes strong mitigation of $CO_2$ emissions, which drop to net zero emissions by 2075 (Gidden et al., 2019). Coupled to the reduction in $CO_2$ emissions, $SO_2$ emissions also decrease significantly in SSP1-2.6, reducing from a global emission of approximately 100 Mtonnes of $SO_2$ per year in 2020 to less
than 20 Mtonnes of $SO_2$ per year in 2070. SSP1-2.6 includes significant land use change, including an increase in global forest cover. In contrast, SSP3-7.0 represents a medium to high end future forcing pathway with $CO_2$ emissions approximately doubling from 2015 to 2100. In addition, SSP3-7.0 assumes a significant increase in methane ($CH_4$) emissions over the coming century, increasing from approximately 400 Mtonnes of $CH_4$ per year in 2020 to slightly less than 800 Mtonnes of $CH_4$ per year in 2100. $SO_2$ emissions do not decrease anywhere near as rapidly in SSP3-7.0, with emissions at 90 Mtonnes of $SO_2$ per
year in 2070 (Gidden et al., 2019).

The forcing data used in all simulations is the same as used in the original UKESM1 CMIP6 simulations and described in detail in Sellar et al. (2020, 2019) and we refer the reader to these papers for more detail.

In total the UKESM1 historical ensemble comprises 19 members. For a more valid comparison of the two models we chose 6 members from UKESM1 to compare against the 6 members of UKESM1.1. Sellar et al. (2019) outline the approach
used to select the initial state of each historical member by sampling different parts of the phase space of the model's multi-





decadal variability, identified by two key modes of variability, the Inter-decadal Pacific Oscillation (IPO) and the Atlantic Multi-decadal Oscillation (AMO). We do not use the same approach in UKESM1.1, instead taking initial conditions from the piControl which are equally spaced 40 years apart. Following this, we choose 6 members of the UKESM1 historical ensemble which are separated by a similar number of years. Details of the ensemble members used as well as the branching dates from

the piControl are given in the Appendix.

## 4 Results

### 4.1 Pre-industrial climate stability

The temporal stability of the UKESM1.1 piControl is shown in Figure 1 which compares the timeseries of a number of key properties of the pre-industrial climate in UKESM1.1 and UKESM1. The net global mean TOA radiation is comparable

between the models with both having a mean net TOA radiation over the 400 year period of approximately $0.04\,\mathrm{W\,m^{-2}}$ with negligible drift. Both models also show very similar interannual variability.

  A key difference in the PI states is the warmer climate of UKESM1.1 which, in the global mean, is up to $0.75\,\mathrm{K}$ warmer at the surface than UKESM1. This leads to less sea ice in both the Arctic and Antarctic with the Arctic showing a larger reduction, with annual mean area reducing from about $12.5\,\mathrm{million\,km^2}$ to $11\,\mathrm{million\,km^2}$. Vegetation fractions are also very stable with

negligible changes in grass and bare soil fractions between UKESM1 and UKESM1.1. There is an increase in tree fraction at the expense of shrubs. This occurs predominantly in the NH high latitudes and is consistent with a warmer climate enabling a slight northward extension of the boreal tree line.

  Any drift in carbon uptake is well within the allowable threshold of $\pm 0.1\,\mathrm{Gt[C]\,year^{-1}}$ which was a requirement for participation in the Coupled Climate-Carbon Cycle Model Intercomparison Project (C4MIP, Jones et al. (2016)). The UKESM1

piControl plotted in Figure 1 is from a much later period than that used in Sellar et al. (2019) and therefore the drift is noticeably smaller here (see their Fig. 3). The ocean carbon uptake in UKESM1.1 appears to oscillate around zero while there appears to be more variability in the soil carbon. In particular after 300 years there is a sudden increase in the release of soil carbon to the atmosphere.

  The variability of global ocean heat content (OHC) in the two pre-industrial simulations (Fig. 2) shows significant multi-

decadal variability in both simulations whereas the trends are very small (in the order of 1 ZJ per century). As part of this inherent variability, in UKESM1.1 there is a phase of increase in OHC (heat gain for the ocean) from year 250 to year 330. There is a concomitant decrease in Southern Ocean sea ice cover (middle panel in Fig. 1), suggesting a temporary warm spell in the Southern Ocean together with changes in the regional deep-water formation processes. Sellar et al. (2019), in their Fig. 3, show a comparable centennial variability in a different part of the UKESM1 pre-industrial control simulation.

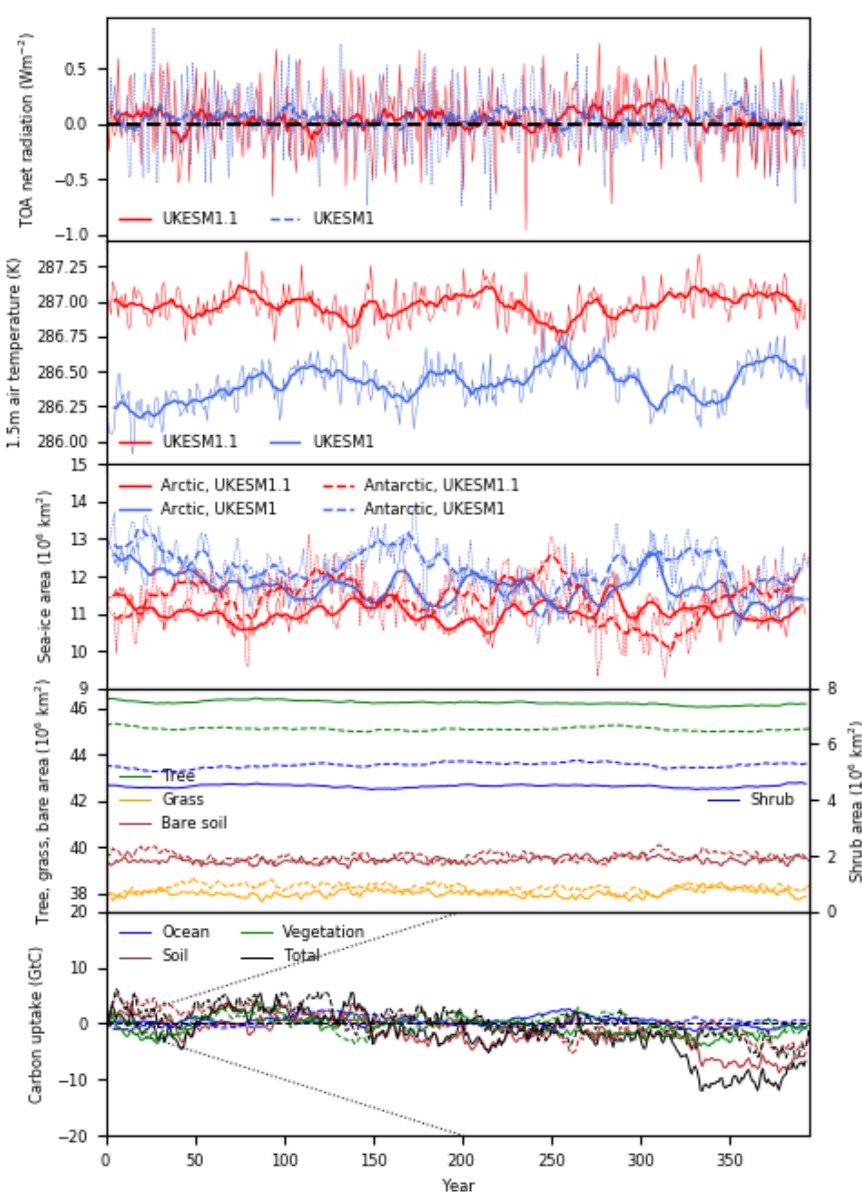

**Figure 1.** Timeseries of pre-industrial climate properties from UKESM1 and UKESM1.1. Plotted from top to bottom are: global annual mean net top of atmosphere radiation; global annual mean 1.5 m air temperature; annual mean Arctic and Antarctic sea ice area; global total area of aggregated plant functional types (solid lines=UKESM1.1, dashed lines=UKESM1); cumulative carbon uptake (solid lines=UKESM1.1, dashed lines=UKESM1, dashed black line depicts acceptable drift following Jones et al. (2016)). In the top 3 panels UKESM1.1 is plotted in red and UKESM1 is plotted in blue. An 11 year running mean is plotted in the thicker lines.

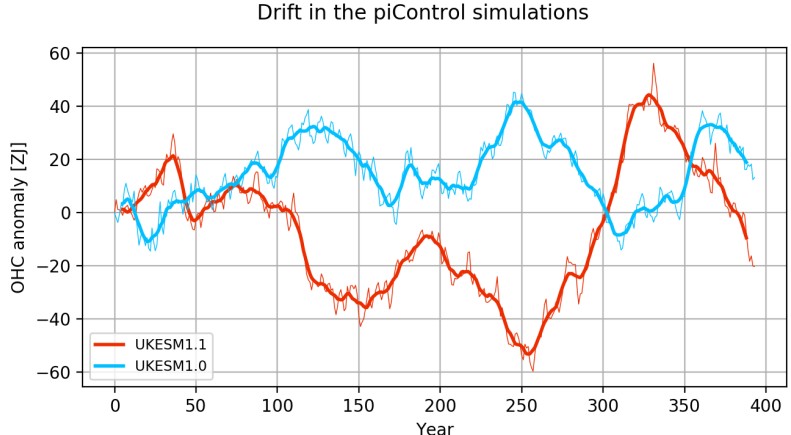

**Figure 2.** Timeseries of pre-industrial ocean heat content anomalies from UKESM1 (red) and UKESM1.1 (blue). Anomalies are calculated relative to the first year of the piControl simulations and an 11-year running mean plotted as in Fig. 1.

## 4.2 Evolution of historical climate

### 4.2.1 Surface temperature

Figure 3 shows the timeseries of global mean historical surface air temperature anomalies of UKESM1 and UKESM1.1 along with observations from the HadCRUT5 (Morice et al., 2021) and Cowtan and Way (Cowtan and Way, 2014) observational datasets. The surface temperature anomalies are calculated relative to the 1850-1900 mean and, even though the drift in the piControl is negligible (see Sec. 4.1), the historical timeseries is nonetheless detrended using the linear trend of the piControl over the equivalent PI period.

The observations (Figure 3a) depict a global warming of approximately 0.5K during the period 1900 to 1940 before cooling slightly in the years following the Second World War, and then warming again from the mid-1970s onwards. UKESM1 shows limited warming between 1900 and 1940 followed by excessive cooling beginning in the late 1950s. The model warms during the 1970 to 2014 period but the rate of warming is overly strong.

The global mean temperature anomaly is generally more positive in UKESM1.1 than UKESM1 throughout the historical period. The two models start to diverge from about 1920, a period coincident with notable increases in anthropogenic $SO_2$ emissions (see Supplementary Figure S3). While still underestimating the strength of the warming during the early part of the century a significant improvement is seen in the anomaly bias against both sets of observations. In particular, the large cold bias from 1960 to the late 1980s is significantly improved in UKESM1.1 with the ensemble mean bias improved by over 50%. Both models cool strongly in response to the Mount Pinatubo volcanic eruption in 1991 after which they both tend to warm at the same rate and UKESM1.1 is slightly too warm by the end of the historical period.

Figure 3 also plots the surface temperature anomalies for the extra-tropical northern (NHex) and southern hemispheres (SHex) as well as for the tropics. It is evident that the discrepancies between the model and observations are predominantly

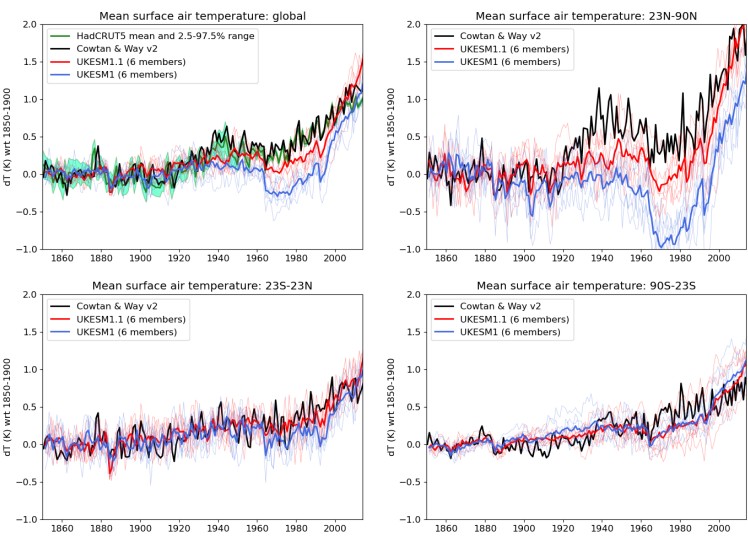

**Figure 3.** Annual mean surface temperature anomaly relative to 1850-1900 mean for the full globe, northern hemisphere extra-tropics (23-90N), tropics (23S-23N) and southern hemisphere extra-tropics (90S-23S). Thin lines are the individual historical ensemble members of UKESM1.1 (red) and UKESM1 (blue); the thick line is their ensemble mean. The Cowtan and Way (2014) temperature reconstructions are plotted in black and, for the full globe, the HadCRUT.5.0.1.0 mean and 2.5-97.5 % uncertainty range (Morice et al., 2021) are shown in dark green and aquamarine. Model time series are de-trended using the linear trend of the pre-industrial control over the equivalent period.

occurring in the NHex which lends support to the likely leading role of anthropogenic aerosols in this bias. In general, the individual ensemble members of both models overlap the observations in both the tropics and SHex. In the SHex, the UKESM1.1 ensemble mean is marginally colder than that for UKESM1, although there is large variability across the ensemble members in this region. While both the tropics and SHex depict a certain amount of inter-annual variability they both show a general warming trend during the historical period. In contrast, the different global warming and cooling periods discussed above are

clearly originating in the NHex region pointing to an anthropogenically-forced influence. The main impact of UKESM1.1 on the surface temperature is also found to occur predominantly in the NHex. This is where the $SO_2$ deposition changes are expected to have maximum effect due to the majority of anthropogenic $SO_2$ being emitted in this region. The model now captures a slight warming during the 1920 to 1940 period although still underestimating the magnitude of the warming seen in the observations. The magnitude of the cold bias in the NHex in UKESM1 approaches 1 K around 1970 while in UKESM1.1

it is reduced to 0.2-0.3 K.

Figure 4 shows the spatial surface temperature trends in UKESM1 and the difference in trends between UKESM1 and UKESM1.1 for the three distinct anomaly periods (i.e. period of warming (1900-1940), cooling (1941-1980) and warming (1981-2014)) already highlighted above. The surface temperature trends are compared to HadCRUT5 observations (Morice



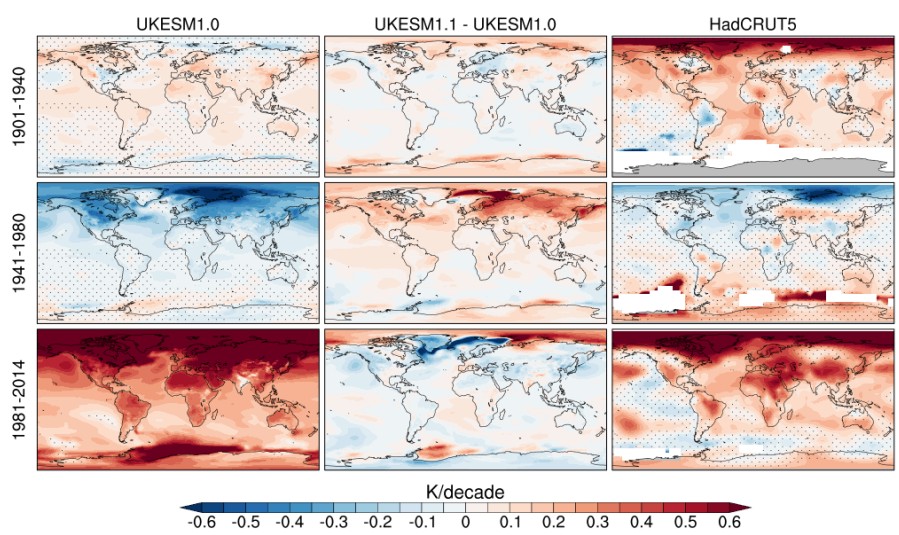

**Figure 4.** Surface temperature trends in (left) UKESM1 and (right) HadCRUT5 (Morice et al., 2021) and (middle) the UKESM1.1-UKESM1 difference for the periods (top) 1901-1940, (middle) 1941-1980 and (bottom) 1981-2014. Regions where trends are not significant are stippled.

et al., 2021). Between 1900 and 1940, the strongest warming trends are observed in the Arctic, which warms in excess of 0.6

K/decade during this period. Warming trends in excess of 0.2 K/decade are also observed over most ocean basins and NH continents. Changes in surface temperature trends in UKESM1.1 relative to UKESM1 are small during this period and the models generally underestimate the magnitude of the warming trends globally. The models also fail to capture the localised regions of continental cooling. The largest change in surface temperature trend between UKESM1.1 and UKESM1 is found between 1940 and 1980. The trends are more positive in UKESM1.1 during this period, with the largest changes occurring

across the northern hemisphere (NH) high latitude regions where the model trends are warmer by up to 0.4 K/decade. This significantly improves the negative bias in the temperature trends during this period. Warming trends observed over central Asia and in the southern hemisphere (SH) high latitudes are not captured well in the models although these trends appear not to be statistically significant. The sign of the temperature trends changes from a predominantly cooling trend in the 1941-1980





period to a predominantly warming trend between 1981 and 2014. This change coincides with the time period where globally

averaged anthropogenic $SO_2$ emissions peak before stabilising and subsequently declining, following the implementation of clean air legislation (see Supplementary Figure S3). During the 1981-2014 period both models warm too fast everywhere but this strong warming trend is generally reduced in UKESM1.1. This improves agreement with the observations particularly in the NH although the more positive trends in the Arctic region further degrade the model bias here. Failure to capture the observed cooling trend over the Pacific Ocean is another common feature across CMIP6 models (Dittus et al., 2021). Several

causes for this possible bias have been proposed, including errors in the internal variability, biased responses to volcanic eruptions or the circulation response to anthropogenic aerosols, among others.

The weaker warming trends off the western coasts of North and South America improves the model bias in these regions but biases persist in the tropical Pacific.

### 4.2.2 Aerosols and clouds

The main science update which has been implemented into UKESM1.1 is the updated $SO_2$ dry deposition parameterization which alters the simulated trace gas $SO_2$ concentrations and subsequent $SO_4$ aerosol formation. In order to investigate the possible drivers of the improved historical surface temperature record presented above, here we examine the change in the historical evolution of key aerosol and cloud properties and how this alters the SW radiative fluxes.

Hardacre et al. (2021) conduct an in-depth evaluation of the impact of the updated $SO_2$ dry deposition parameterization on

the surface $SO_2$ and $SO_4$ concentrations in UKESM1 using both ground-based and satellite data. The simulations assessed in that study also include the updates to the DMS chemical reactions (see Table 1) and the bugfix for the updating of $H_2SO_4$ concentrations but do not include any of the other changes implemented in UKESM1.1. The increase in $SO_2$ dry deposition in UKESM1.1 reduces the global $SO_2$ gas and $SO_4$ aerosol burdens. Figure 5 compares the surface $SO_2$ and $SO_4$ concentrations with surface measurements across Europe and North America. Full details of the observational data are provided in Hardacre

et al. (2021). Similar to the findings in Hardacre et al. (2021) a notable improvement in the positive surface $SO_2$ concentration bias is found in UKESM1.1. A corresponding reduction in $SO_4$ aerosol concentrations degrades the pre-existing negative bias (Mulcahy et al., 2020; Hardacre et al., 2021). This points to the possibility of further compensating biases in the sulphur cycle with too little $SO_2$ loss via chemical oxidation of $SO_2$ by OH and $O_3$, combined with potentially too large anthropogenic emissions of $SO_2$ being compensated for here by dry deposition.

Simulated anomalies of the annual mean AOD, cloud droplet number concentration ($N_d$) and cloud droplet effective radius ($r_{eff}$) over the historical period are presented for the NHex region in Figure 6. The anomalies are calculated relative to an 1850-1900 mean. Both the AOD and $N_d$ begin to steadily increase early in the 20th century with a more rapid increase occurring from about 1950 for both variables. The increase in $N_d$ is accompanied by a corresponding decrease in the $r_{eff}$. Anomalies in all three variables peak around 1980 before slowly declining.

Clearly the evolution of the aerosol and cloud properties is closely tied to the evolution of the anthropogenic $SO_2$ emissions (Supplementary Fig. S3). While both models show a similar temporal evolution in AOD, $N_d$ and $r_{eff}$ the anomalies in UKESM1.1 are systematically smaller in all cases. The strong increasing trend in AOD and $N_d$ over the 1940 to 1990 period



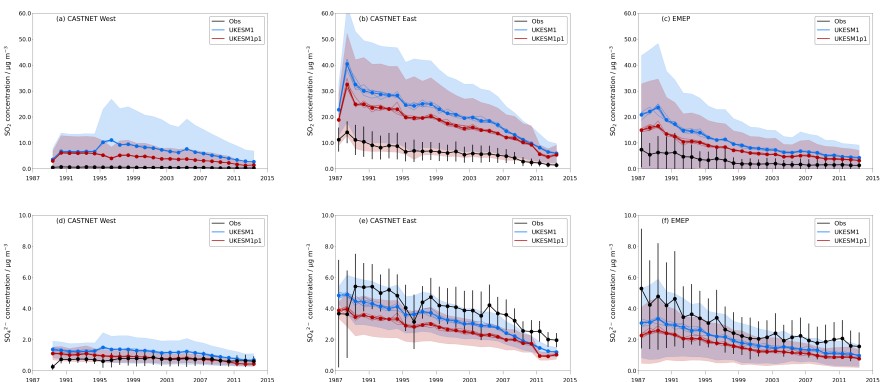

**Figure 5.** Comparison of annual mean (top) surface $SO_2$ concentration and (bottom) surface $SO_4$ aerosol concentration across a network of North American (CASTNET) and European (EMEP) measurement sites. CASTNET stations have been grouped into East and West sites as determined by their location east and west of $100°$ W.

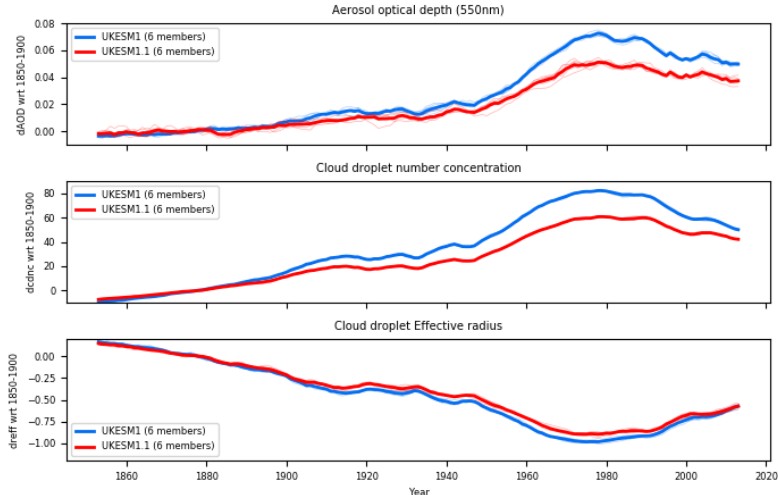

**Figure 6.** Historical evolution of anomalies in aerosol optical depth (550 nm), cloud droplet number concentration and cloud effective radius across the northern hemisphere extra-tropics in UKESM1 (blue) and UKESM1.1 (red). Anomalies are calculated relative to the 1850-1900 mean. Both the ensemble mean (thick lines) and individual ensemble members (fainter lines) are plotted. A 5 year running mean has been applied to the annual mean model output.

simulated by UKESM1 is weaker in UKESM1.1 with peak anomaly values reduced by approximately 30% in both variables and the anomaly in $r_{eff}$ reduced by approximately 10%.

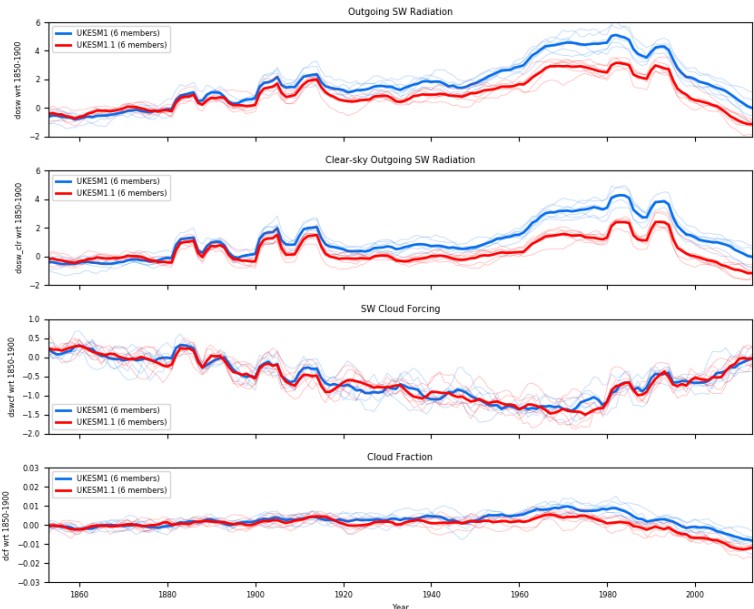

**Figure 7.** Historical evolution of anomalies in all-sky and clear-sky outgoing SW radiation, SW cloud forcing and cloud fraction relative to the 1850-1900 mean in UKESM1 (blue) and UKESM1.1 (red). Both the ensemble mean (thick lines) and individual ensemble members (fainter lines) are plotted. A 5 year running mean has been applied to the annual mean model output.

The reductions in AOD and $N_d$ anomalies in UKESM1.1 relative to UKESM1 drive reductions in the outgoing shortwave (OSW) radiation at the top-of-atmosphere (Figure 7). The anomaly in OSW over the historical period has some large inter-annual variations associated with the large volcanic eruptions. Overall however there is a general increasing trend in OSW throughout the 20th century peaking again at 1980. The positive anomalies in both clear-sky and all-sky OSW are smaller throughout the period in UKESM1.1 by between 30% and 40%. The difference in the SW cloud forcing anomaly is negligible between the two models although there is large variability among the different ensemble members. These findings suggest that the reduced anomaly in the clear-sky OSW radiation are driving the more positive surface temperature anomalies in UKESM1.1.

Lack of historical observations of aerosol and cloud properties preclude an evaluation of the evolution of these properties and their trends over the full historical period. We are therefore reliant on observations of the more recent past to give us an indication of the skill of the models in simulating these properties in the present-day. Such an evaluation will provide useful evidence as to the overall skill of the aerosol simulation in the updated UKESM1.1 configuration.

Annual mean AOD over the 2003 to 2014 period is compared to a number of satellite AOD products in Figure 8. These products, their uncertainties and details of the evaluation procedure are described in detail in Mulcahy et al. (2020). The time



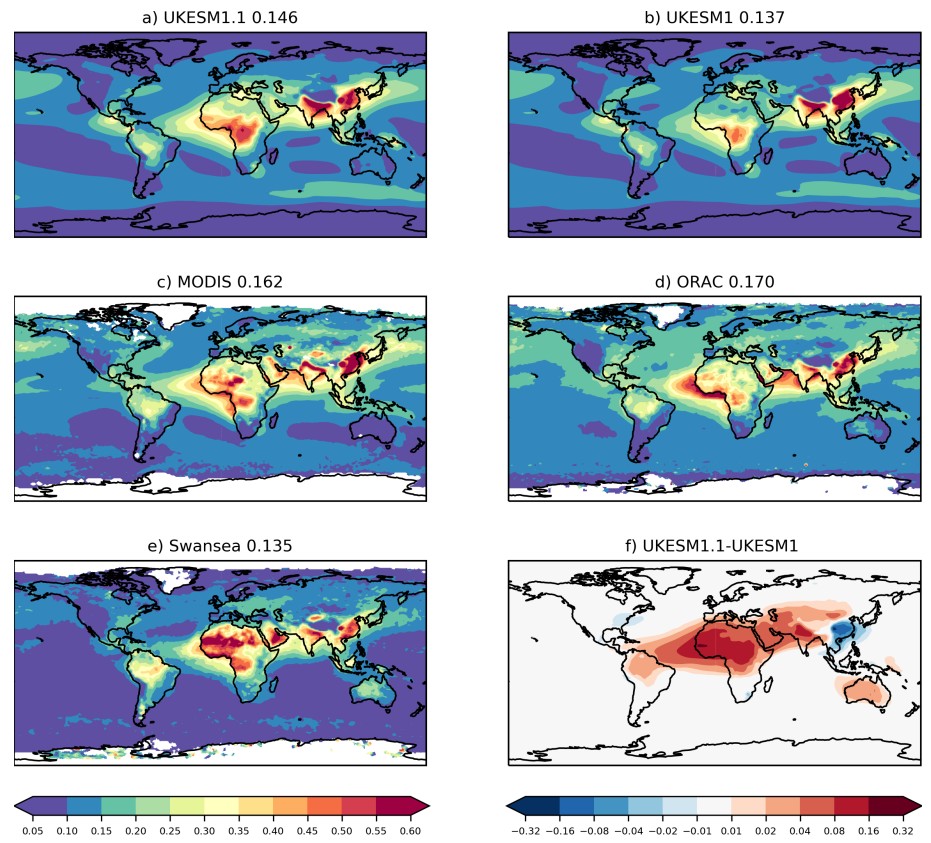

**Figure 8.** Annual mean AOD at 550 nm from (a) UKESM1.1, (b) UKESM1 simulations over the 2003-2014 period; satellite-derived AOD over the same period from (c) MODIS c6 (Hsu et al., 2004), (d) ORAC (Thomas et al., 2009) and (e) Swansea retrieval algorithms (Bevan et al., 2012); (f) UKESM1.1-UKESM1 AOD difference. The global mean AOD from models and satellites are shown in the panel title.

period is chosen to match the available satellite data record. Despite a reduction in the positive anomaly in AOD over the

historical period (Figure 6), the global mean AOD in UKESM1.1 is higher than UKESM1 by about 7%. This is due to the revised dust tuning in UKESM1.1 which increases the dust AOD over and downwind of the major dust source regions. The higher AOD improves low model biases in these regions, in particular the low bias over the Sahara desert and tropical Atlantic as well as the Arabian Sea are in better agreement with the satellite data. Increased AOD over Australia improves agreement with the ORAC and Swansea products but introduces a positive bias against MODIS. Reductions in AOD can be seen in the

important anthropogenic emission source regions of China and northeast America with the reductions in China being the largest due to the much higher $SO_2$ emissions in China during this period. Interestingly, reductions in AOD over Europe are not found. Emissions of $SO_2$ in Europe are much reduced by 2003 and so contributions of $SO_4$ to the total AOD will be relatively smaller during this time. Furthermore, any decrease in AOD due to changes in $SO_4$ load has potentially been offset by a corresponding increase in AOD from dust transported from the Sahara. The reduction in AOD over China improves the large positive bias

in this region and the spatial heterogeneity in the observed AOD over this region is better captured in UKESM1.1. While the





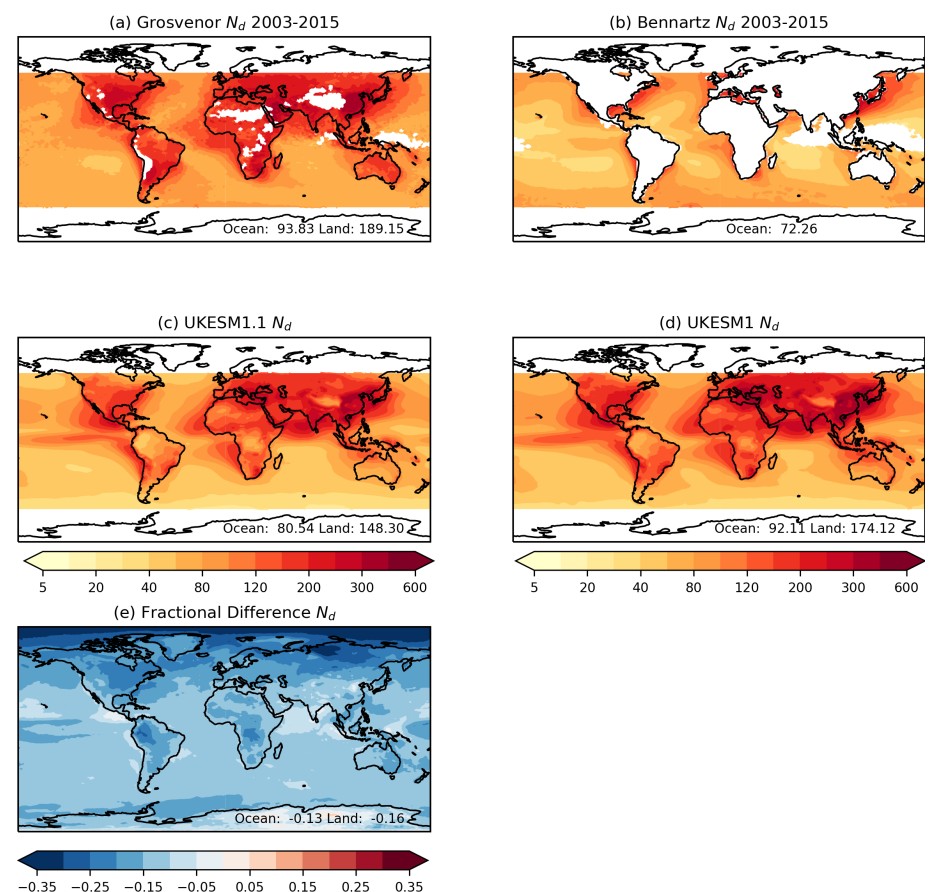

**Figure 9.** Observed and simulated annual mean cloud droplet number concentration, $N_d$ (cm$^{-3}$), from (a) Grosvenor et al. (2018), (b) Bennartz and Rausch (2017) satellite products and simulated $N_d$ from (c) UKESM1.1 and (d) UKESM1; (e) UKESM1.1-UKESM1 $N_d$ fractional difference. The mean ocean and land $N_d$ values are displayed in the sub-panels.

uncertainty in the satellite products can be large, UKESM1 AOD was on the lower end of the satellite AOD range (Mulcahy et al., 2020) and UKESM1.1 now comfortably sits in the middle of this range.

The ground-based AERONET network of AOD measurements (Holben et al., 1998) provides another source of reliable AOD measurements with lower uncertainties than the satellite products (Holben et al., 2001). A comparison of the simulated AOD at 440 nm with a long-term climatology of measurements from 67 AERONET sites is shown in Supplementary Figure S4. Both the root mean square error and correlation coefficient is improved in UKESM1.1. Agreement is generally improved at measurement sites in the Sahel, Saudi Arabia and the Carribean. The AOD gradient from west to east USA also appears to be better captured.

In UKESM mineral dust is treated as hydrophobic and therefore cannot act as cloud condensation nuclei. The global mean increase in AOD is therefore not translated into a corresponding increase in cloud droplet number concentration which will





respond more to the changes in $SO_4$ aerosol in this configuration. There is a global-scale reduction in present-day $N_d$ in UKESM1.1 (Figure 9) with reductions being marginally higher over land (reduced by 16%) than ocean (reduced by 13%) surfaces. Reductions in $N_d$ are much higher in the NH with reductions of approximately 30% found in the NH high latitudes. Again there is large uncertainty in the satellite retrievals of $N_d$ as discussed in Mulcahy et al. (2020) and the spatial distribution
and magnitude of $N_d$ in both model configurations lies within this observational uncertainty. While UKESM1 arguably is closer to the Grosvenor et al. (2018) product, UKESM1.1 now sits within the two satellite estimates.

### 4.2.3 Ocean heat content

Changes in ocean heat content (OHC) account for 91% of the increase in the global energy inventory (Forster et al., 2021). Changes in ocean heat content are therefore a more reliable indicator of Earth's energy imbalance than, for example, sea
surface temperature (von Schuckmann et al., 2016). Here we assess the capability of UKESM1.1, compared against UKESM1, to reproduce the observed increase in OHC during the historical period.

There are various observation-based datasets of historical OHC. Most of them are based on the same in-situ observations of ocean temperature but use different methods for infilling gaps in observations and for correcting instrumental biases (Boyer et al., 2016). The coverage of in-situ observations is sparse below a depth of 2000 m, and before 1960 in the upper layers. We
use a selection of OHC observational datasets to illustrate this observational range: NCEI (Levitus et al., 2012); C2020 (Cheng et al., 2020); EN421 (Good et al., 2013); DOM+LEV (Domingues et al., 2008; Levitus et al., 2012); ISH (Ishii et al., 2017), and CHG (Cheng et al., 2017). For depths below 2000 m we used the OHC trend estimate from Purkey and Johnson (2010). The historical simulations are detrended by diagnosing the linear trend from the matching part of the piControl and subtracting it from the historical simulation. OHC anomalies are calculated relative to a 2005-2014 mean as this period is best covered by
the observations and enables a more realistic model-observation comparison.

In the 0-700 m ocean layer (Figure 10b) the simulations are reasonably close to observations from about 1960 onwards. Prior to 1960 the observations indicate a larger change in OHC than is simulated by the UKESM ensembles. While UKESM1.1 is notably closer to observations than UKESM1, both model versions indicate little change in OHC during the 1970s and 1980s and do not capture the observed increase of 50 ZJ. After 1991 the rate of OHC change in both model versions is stronger than
observed. This is partly a consequence of the relatively large climate sensitivity of these models (see sec. 4.3) responding to the increase in greenhouse gas concentrations in this period. The models appear to react strongly to the 20th century volcano eruptions in 1963 (Agung), 1982 (El Chichon) and 1991 (Pinatubo) (see Figure 10a) with a marked loss in OHC on the order of 10 ZJ or 20 ZJ (Figure 10b). In particular for the Pinatubo eruption in 1991, observations do not seem to indicate such a loss of global OHC (Gregory et al., 2020). The UKESM processes responsible for this strong reaction are currently under
investigation.

For the entire global ocean (Figure 10c) the model performance is different to that of the 0-700 m layer. The performance improvement in UKESM1.1 is much larger for the entire global ocean. The total 1950-2014 OHC increase is about 40% larger in UKESM1.1, bringing it much closer to observations. For the period since 1991 the rate of OHC change is still larger than

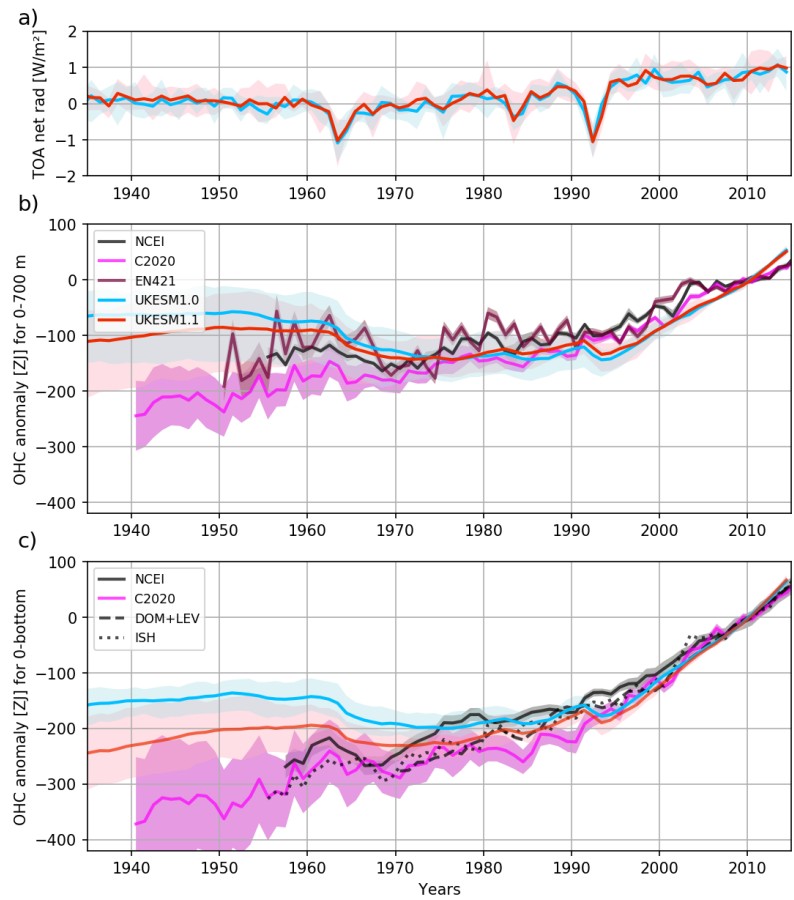

**Figure 10.** (a) TOA net radiation, (b) 0-700 m global ocean heat content (OHC) anomaly and (c) full-depth global OHC anomaly for the UKESM1.1 (red) and UKESM1 (blue) historical simulations. Observation-based OHC time series (b,c) are plotted in black, purple and maroon. TOA net radiation time series are anomalies with respect to the average of 1850-1870. OHC time series (simulations and observations) are anomalies with respect to the average of 2005-2014. Shading shows the range of the ensemble for simulations, and the standard error for the observational data.

observations, but only marginally so. This implies that in the ocean layers below 700 m the uptake of heat is small during this
period, compensating for the strong increase above 700 m.

A regional breakdown of OHC changes since 1971 is given in Figure 11. The ocean basins employed are the Atlantic, the Indian and the Pacific, each separated at the equator into a Northern and a Southern section. We analyse ocean heat uptake (OHU, defined as the change in OHC) for two consecutive periods. The period 1971-1991 represents a period of small ocean



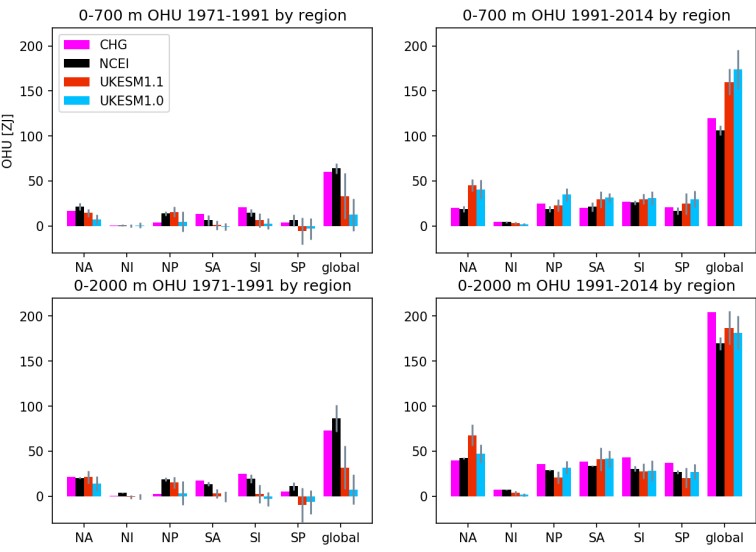

**Figure 11.** Ocean heat uptake (OHU) for six ocean basins and the global ocean in the 0-700 m layer (top) and the 0-2000 m layer (bottom) for the 1971-1991 period (left) and the 1991-2014 period (right). Colour code as in Fig. 10. Thin grey bars give the standard error of the observational data sets and the ensemble standard deviation for the model ensembles. The ocean basins included are North and South Atlantic (NA, SA), North and South Indian (NI, SI), North and South Pacific (NP, SP).

heat uptake, while in the period 1991-2014 the OHU is much larger. We used the 0-2000 m layer here instead of the full-depth ocean because no OHC trend estimates for individual basins are available below 2000 m. For 1971-1991 (Figure 11) we see a clear improvement in UKESM1.1, on average capturing about half of the observed OHU (in 0-700 m). This increase comes mainly from the North Atlantic (NA) and North Pacific (NP) basins responding to the less negative anthropogenic aerosol ERF in UKESM1.1 (see Sec 4.3). In contrast to observations we see negative OHU (loss of heat from the ocean) in the south Indian (SI) and south Pacific (SP) basins. We suggest that this is heat loss from simulated open-ocean deep water formation (Menary et al., 2018), which does not correctly capture the actual deep water formation processes on the Antarctic shelves.

In the 1991-2014 period we see again that both model versions simulate too much OHU in the 0-700 m layer. The source of this excess heat is mainly the North Atlantic, and to some extent the South Atlantic as well. Kuhlbrodt et al. (2018) (see their Fig. 8) showed a similar strong warming trend in the deep Atlantic in HadGEM3-GC3.1-LL, the physical core model of UKESM1 and UKESM1.1. It is possible that this tendency for strong deep Atlantic OHU is the underlying reason here.

### 4.2.4 Ocean circulation

Yool et al. (2021) highlight a number of biases in the interior ocean circulation in UKESM1, such as insufficient vertical mixing leading to surface heat biases and dampened deep water circulations. Here we examine the impact of the different



atmospheric forcing in UKESM1.1, driven primarily by weaker anthropogenic aerosol forcing, on these biases. Figure 12
shows the global streamfunction of the ocean's meridional overturning circulation (MOC), produced by the Estimating the

Circulation and Climate of the Ocean consortium (ECCO, (Forget et al., 2015; Fukumori et al., 2021)) ocean reanalysis and
compares it with UKESM1 and UKESM1.1. Both model MOCs are broadly consistent with the ECCO reanalysis. Both exhibit
a stronger maximum MOC at $40\,^\circ$N, and a slightly weaker Antarctica Bottom Water (AABW) cell northward of $60\,^\circ$S. The
maximum strength of the AABW cell in UKESM1.1 is at $45\,^\circ$S and is slightly weaker than that of UKESM1, but otherwise
the models perform very similarly.

On a regional scale, Figure 13 shows the complete historical timeseries for the Atlantic MOC (AMOC) and Drake Passage
transport. The observational mean for the AMOC for the period 2004-2014 is $16.8\,\mathrm{Sv}$ (Smeed et al., 2018), compared with sim-
ulated magnitudes of 16.4 and $15.7\,\mathrm{Sv}$ for UKESM1 and UKESM1.1 respectively. Driven by anthropogenic aerosol-mediated
cooling of the northern hemisphere (Menary et al., 2020), the AMOC strength exhibits a "ramping-up" throughout the 20th
century, followed by a decline into the 21st. The slightly lower AMOC in the UKESM1.1 ensemble mean is consistent with a

weaker aerosol forcing in the UKESM1.1 simulations, although there is considerable variability across the ensemble members
leading to the configurations largely overlapping particularly in the first half of the simulated period (1850 to approximately
1940). The Drake Passage transport is estimated at $173\,\mathrm{Sv}$ (Donohoe et al., 2016), with simulated magnitudes of 157.5 and
$156.7\,\mathrm{Sv}$ for UKESM1 and UKESM1.1 respectively (for the period 2004-2014). UKESM1.1 shows very similar magnitudes
to UKESM1 for both major transports, if slightly lower on average for both. The temporal patterns of both transports across

the historical period are also very similar.

### 4.2.5 Sea ice

The cold bias in UKESM1 drives positive biases in Arctic sea-ice extent and thickness (Sellar et al., 2019; Yool et al., 2021).
Here we examine what impact the warmer climate in UKESM1.1 has on sea-ice properties in both the Arctic and Antarctic.
Previous studies have highlighted a strong negative correlation between global mean surface temperature and Arctic sea ice

extent (Rosenblum and Eisenman, 2017; Mahlstein and Knutti, 2012). Figure 14 shows the historical timeseries of sea ice extent
in the Arctic for the months corresponding to periods of maximum (March) and minimum (September) sea ice in this region.
In both seasons the warmer climate in UKESM1.1 leads to a reduction in sea ice extent and volume throughout the historical
period. Observational estimates from the HadISST (HadISST.2.2.0.0; (Titchner and Rayner, 2014)) and NSIDC (Fetterer et al.,
2017) datasets available from 1979 onwards are also plotted. While UKESM1 agrees better in sea ice extent with the HadISST

data in March it is positively biased against NSIDC. UKESM1.1 therefore moves further away from the HadISST data but
agrees well with NSIDC. In September, the observational data are in better agreement with each other and with UKESM1.1
while UKESM1 is positively biased against both. The trend in observed Arctic sea ice extent (and volume) is negative over the
observed period, with a stronger negative trend evident during sea ice minimum periods. Both models also simulate decreasing
trends during this period. However, in UKESM1 the trend is too strong in March indicating that the model simulates a faster

rate of loss of sea ice in this region than observed. Meanwhile the very high ice thickness in the central Arctic means that the



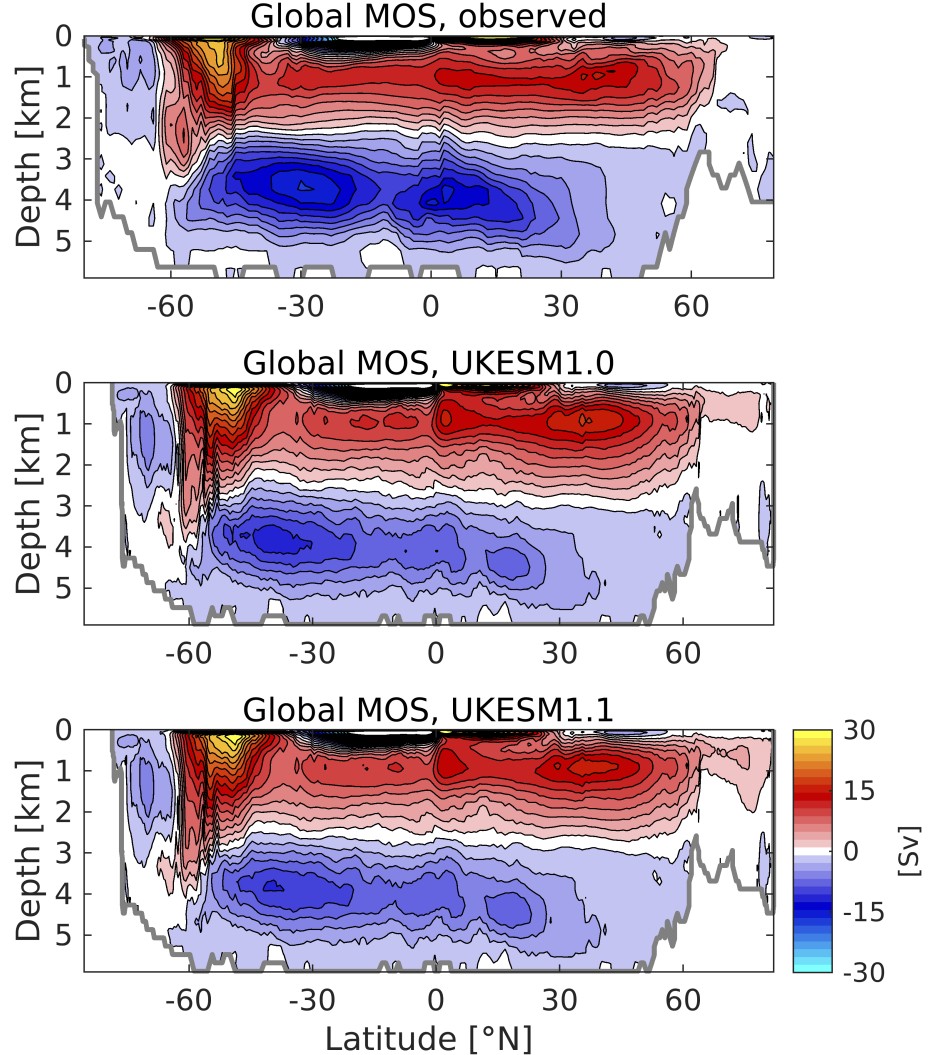

**Figure 12.** Observationally-derived (top) and UKESM1 (middle) and UKESM1.1 (bottom) meridional overturning circulation (MOC) for the global ocean . The observational circulation is derived from the ECCO V4r4 ocean circulation reanalysis for the period 1992-2017. The model circulation shown is based on the decadal mean streamfunction for the period 2000-2009, averaged across the members of both model ensembles. Both plots include the components from parameterised mesoscaled eddies (Gent and McWilliams, 1990; Gent et al., 1995). MOC is in Sv with a contour interval of 2 Sv.

UKESM1 trend in September is too weak. In UKESM1.1 the trends are in better agreement with the observed trends during both periods.

Figure 15 shows the historical timeseries of the Arctic sea ice volume for the same months. Large positive biases in Arctic sea ice volume found in UKESM1 are significantly improved in UKESM1.1 in comparison with the commonly used Pan-Arctic 490 Ice Ocean Modeling and Assimilation System reanalysis dataset (PIOMAS, Schweiger et al. (2011)). UKESM1 simulates a



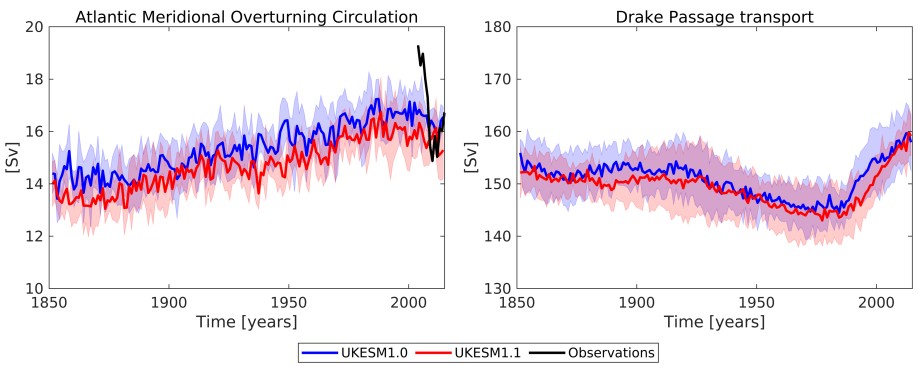

**Figure 13.** Atlantic Meridional Overturning Circulation (AMOC; left) and Drake Passage transport (right) for the UKESM1 (blue) and UKESM1.1 (red) ensembles. Solid lines denote the ensemble mean, with shaded areas marking ±1 standard deviation. Estimated AMOC transport derived from the RAPID array (Smeed et al., 2018) is shown in black for the period of available observations.

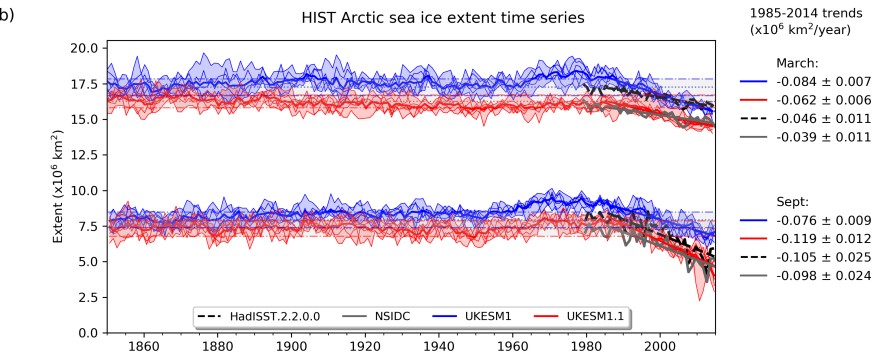

**Figure 14.** Timeseries of simulated (blue, UKESM1; red, UKESM1.1) and observed (black, HadISST; grey, NSIDC) sea ice extent in the Arctic for the months of March and September over the full historical period (1850-2014). The seasonal trends in the more recent period are shown for each dataset on the right-hand side. Model data represents the ensemble mean for UKESM1 (6 members) and UKESM1.1 (6 members) with ±1 standard deviation also shown.

positive trend in sea ice volume throughout the historical period with a notable increase occurring around 1950, reaching peak volume in 1980 before declining steadily to the present day. By contrast, UKESM1.1 shows a minimal trend in the ensemble mean up to 1950, followed by a much smaller increasing trend after this before declining at a similar rate to UKESM1. UKESM1.1 is in better agreement with the PIOMAS dataset during the reanalysis period post-1975 reaching comparable present-day volumes. However both models decrease too sharply after 1980. Consistent with the minimal changes in Southern Hemisphere temperature, there is no significant difference in Antarctic sea ice extent or volume between UKESM1 and UKESM1.1 (see Supplementary Figures S5 and S6).

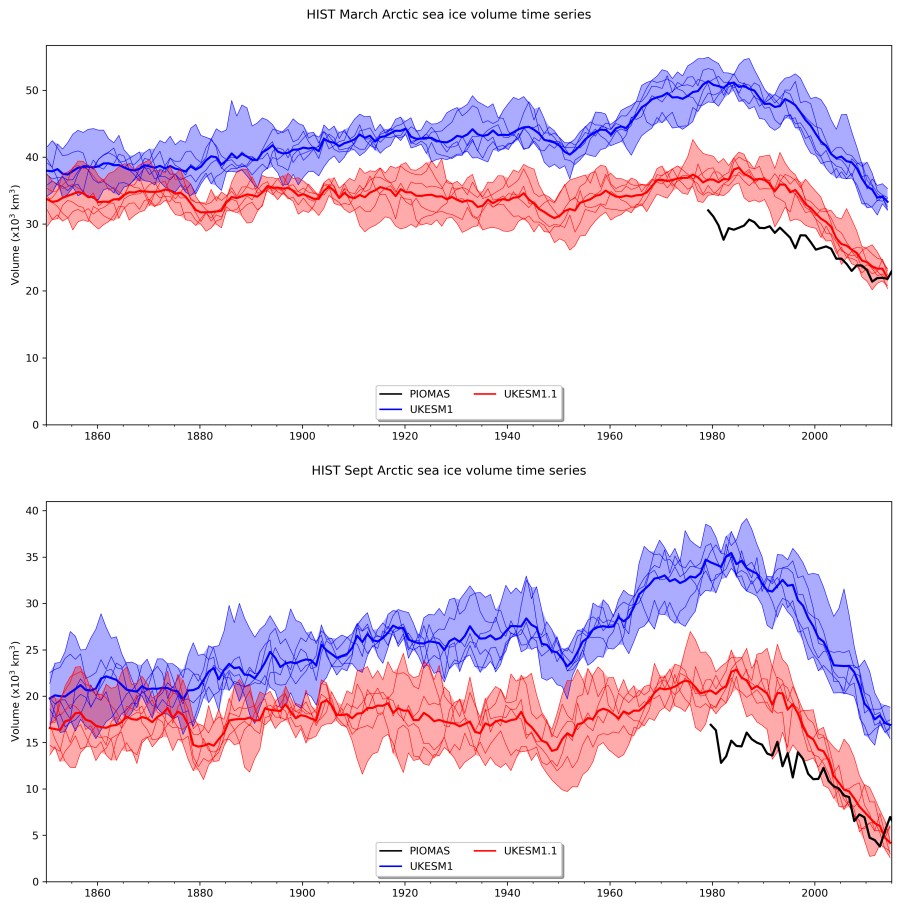

**Figure 15.** Timeseries of simulated (blue, UKESM1; red, UKESM1.1) sea ice volume in the Arctic for the months of (top) March and (bottom) September over the full historical period (1850-2014) along with the PIOMAS reanalysis data (black). Model data represents the ensemble mean for UKESM1 (6 members) and UKESM1.1 (6 members) with ±1 standard deviation also shown.

### 4.2.6 Ocean biogeochemistry

The ocean biogeochemistry model in UKESM1.1 remains unchanged from UKESM1 and changes in the marine biogeochem-
ical properties are therefore small and are solely driven by the changes in the physical climate and coupled interactions such as the iron deposition to the ocean. Supplementary Figure S7 demonstrates the strong agreement in both the magnitude and temporal pattern of the ocean $CO_2$ uptake over the historical period. The total integrated $CO_2$ uptake over this period differs by less than $0.1 \, \mathrm{Pg\,C}$ between UKESM1 and UKESM1.1.

The global mean net ocean primary production (NPP) is also very similar in both models across the historical period,
although UKESM1.1 ($46.6 \, \mathrm{Pg\,C/yr}$) is consistently lower than UKESM1 ($47.9 \, \mathrm{Pg\,C/yr}$) (Figure S7). Changes in the spatial distribution of NPP are also very small (Figure 16) and highlight common regional biases such as the positive bias in the





**Table 3.** Global annual mean effective radiative forcings ($W\,m^{-2}$) in UKESM1.1 and UKESM1. UKESM1 values are taken from O'Connor et al. (2021). The ERFs are calculated for the year 2014 relative to an 1850 pre-industrial control climate (Pincus et al., 2016)

| Forcing | UKESM1 | UKESM1.1 |
|---------|--------|----------|
| Total anthropogenic | 1.76 | 1.84 |
| Well-mixed GHGs | 2.91 | 2.84 |
| Aerosol | -1.09 | -1.01 |
| Land Use | -0.17 | -0.22 |

Southern Ocean and negative biases in the sub-equatorial ocean basins. Changes in surface nitrogen nutrient biases relative to the observed climatology of Garcia et al. (2013) are more noticeable, with lower positive biases in the Southern Ocean and, particularly, the equatorial Pacific. In examining UKESM1, Yool et al. (2021) attribute this latter bias in part to more extreme iron stress in this region caused by low biases in dust deposition from the atmosphere. The patterns of this deposition in UKESM1 and UKESM1.1 (Figure 16) both differ from that of the observational estimate of Mahowald et al. (2005). In particular, fluxes are too small in the equatorial Pacific and Southern Ocean in both models. These biases are generally improved in UKESM1.1 due to the shift in atmospheric dust size distribution to smaller particles leading to more transport of dust over the remote oceans and therefore enhanced oceanic deposition. However, increases in dust deposition across the maritime continent and around Australia increase positive biases in these regions. In general, the higher dust deposition to the ocean surface reduces iron stress and simultaneously increases nitrogen uptake and loss from the surface, reducing surface concentration biases (Figure 16). On balance this results in the model being slightly more nutrient-stressed at the surface, reducing productivity, but the overall changes in net primary productivity are small and do not noticeably impact the bias.

### 4.3 Forcing and climate sensitivity

As noted earlier, the warmer climate in UKESM1.1 and improved representation of the historical surface temperature is believed to be largely due to a weaker anthropogenic aerosol forcing driven predominantly by lower sulphur dioxide and hence sulphate aerosol burdens in the updated configuration. Figure 17 compares the all-sky total anthropogenic, aerosol and land use ERF from UKESM1 and UKESM1.1 and the global mean values are summarized in Table 3. The total anthropogenic ERF in UKESM1.1 has increased by $0.08\,W\,m^{-2}$ with this change being driven almost entirely by a weaker aerosol ERF which has increased from $-1.09\,W\,m^{-2}$ to $-1.01\,W\,m^{-2}$. Most of the change in the aerosol ERF comes through changes in the clear-sky forcing which has increased by $0.2\,W\,m^{-2}$ (see Supplementary Figure S8). This implies that the aerosol-cloud forcing in UKESM1.1 is stronger than UKESM1 by about $0.1\,W\,m^{-2}$. The increases in the aerosol ERF are seen predominantly across the NH although there are also regions of more positive aerosol forcing across the Southern Ocean possibly due to changes in the background natural loading of $SO_4$ aerosol derived from marine DMS sources.

Using the *piClim-control* and *piClim-aer* experiments we further decompose the aerosol ERF into its SW components using the Approximate Partial Radiative Perturbation (APRP) technique (Zelinka et al., 2014; Taylor et al., 2007). Given that

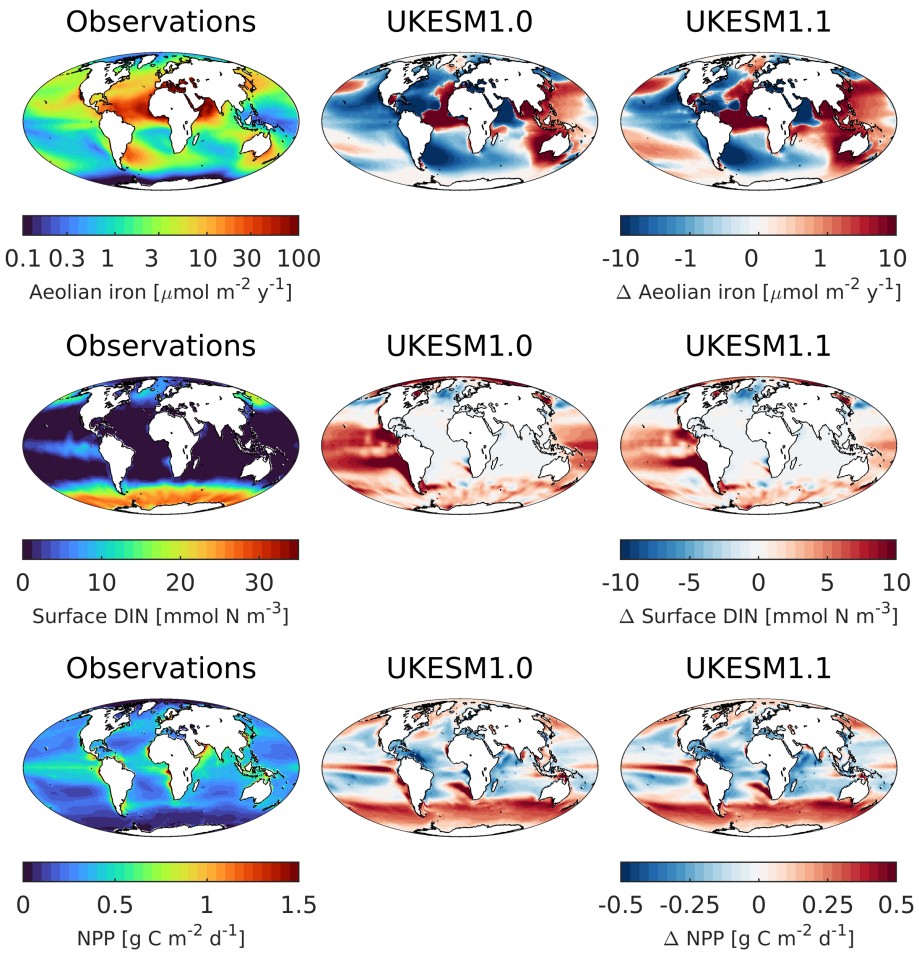

**Figure 16.** Spatial distributions of (left) observed (top) iron deposition to the ocean, (middle) dissolved inorganic nitrogen (DIN) and (bottom) net ocean primary productivity (NPP) along with simulated biases from (middle) UKESM1 and (right) UKESM1.1.

aerosols predominantly perturb the SW part of the spectrum this enables an estimation of the aerosol ERF due to aerosol-radiation interactions (ARI) and aerosol-cloud interactions (ACI) separately. The results are summarized in Table 4. The net SW forcing has increased by $0.13 \, \mathrm{W \, m^{-2}}$. This weaker aerosol forcing is almost entirely due to smaller (more positive) clear-sky scattering component which is offset somewhat by a small strengthening in the SW cloud component. The most notable change in the latter appears to come from an increase in the cloud amount contribution. However, the net SW cloud differences due to aerosols are relatively small ($-0.04 \, \mathrm{W \, m^{-2}}$) overall.

The $SO_2$ dry deposition changes not only reduce the $SO_2$ and subsequently $SO_4$ aerosol concentrations in the *piClim-aer* simulation but also change the pre-industrial aerosol concentration through reductions in $SO_4$ aerosol derived from natural marine emissions of DMS, thus essentially leading to a cleaner PI atmosphere. In addition smaller changes to the overall $SO_4$ burden and natural background $N_d$ are expected from the DMS chemical reaction changes as well as the bugfix in the aerosol

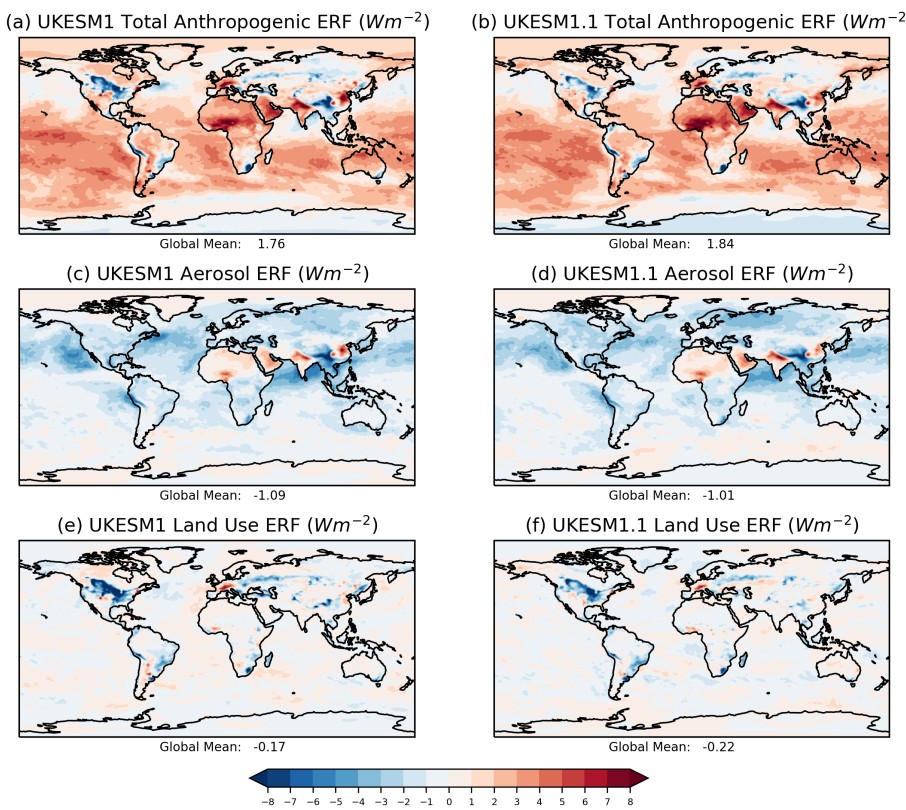

**Figure 17.** Comparison of the anthropogenic effective radiative forcing (ERF) in (left) UKESM1 and (right) UKESM1.1; (top) the total anthropogenic ERF, (middle) aerosol ERF and (bottom) land use change ERF. The ERFs are calculated for the year 2014 relative to an 1850 pre-industrial control following Pincus et al. (2016).

**Table 4.** The SW components of the aerosol effective radiative forcing (W m$^{-2}$) in UKESM1.1 and UKESM1.

| Forcing | UKESM1 | UKESM1.1 |
|---|---|---|
| Aerosol-radiation interactions | | |
| Scattering | -0.75 | -0.60 |
| Absorption | 0.48 | 0.50 |
| Net | -0.27 | -0.10 |
| Aerosol-cloud interactions | | |
| Scattering | -0.85 | -0.86 |
| Absorption | 0.003 | 0.003 |
| Cloud Amount | -0.10 | -0.12 |
| Net | -0.95 | -0.99 |





activation (see Sec. 2.1) which leads to lower pre-industrial $N_d$ concentrations (reduced by 14%) in UKESM1.1. A cleaner pre-industrial climate will strengthen the magnitude of the anthropogenic aerosol effective radiative forcing (ERF) from aerosol-cloud interactions (Carslaw et al., 2013). At the same time the pre-industrial clear-sky AOD is 8% higher in UKESM1.1 due to

the increased dust concentrations which offsets the global mean AOD reduction due to the reduced sulphate aerosol load. This combines with the lower anthropogenic $SO_4$ aerosol to weaken the clear-sky aerosol ERF (see Supplementary Figure S8).

The land use ERF (Figure 17e and f) in UKESM1.1 has strengthened from -0.17 $\mathrm{W\,m^{-2}}$ to -0.22 $\mathrm{W\,m^{-2}}$ while the clear-sky component has increased from -0.28 $\mathrm{W\,m^{-2}}$ to -0.20 $\mathrm{W\,m^{-2}}$ in response to the retuning of the snow burial of vegetation. The cloud masking of the surface albedo change in *piClim-LU* is clearly weaker in UKESM1.1.

The *abrupt4xCO2* and *1%CO2* simulations are used to calculate the transient climate response (TCR) and effective climate sensitivity (EffCS) using the methodology described in Andrews et al. (2019). As reported by Andrews et al. (2019) the TCR (2.76 K) and EffCS (5.36 K) of UKESM1 are both outside of the CMIP5 5-95% ranges. Figure 18 compares the surface air temperature change in the *abrupt4xCO2* and *1%CO2* simulations of UKESM1 and UKESM1.1. We note that four ensemble members were run with UKESM1 while only one simulation was run with UKESM1.1 for each experiment. The surface

temperature response of the two models is almost identical at least at the global scale. This leads to values of EffCS (5.27 K) and TCR (2.64 K) for UKESM1.1 which are very similar to those for UKESM1.

Warmer temperatures in response to enhanced $CO_2$ can enhance the emission of natural aerosol sources, such as DMS which is then oxidised to $SO_2$ and subsequently forms $SO_4$ aerosol. Bodas-Salcedo et al. (2019) found that the introduction of the GLOMAP-mode aerosol scheme into the HadGEM3-GC3.1 and UKESM1 models reduces a negative cloud feedback,

via a weaker DMS-climate feedback in the Southern Ocean, and this partly contributes to the enhanced climate sensitivity in these models. While the pre-industrial background $SO_4$ aerosol will be different in UKESM1.1 the dry deposition and DMS chemistry changes won't necessarily change the DMS emission response in the warmer climate.

Andrews et al. (2019) found a notable impact of the biogeochemical feedback from vegetation (via the $CO_2$ fertilisation effect) on the simulated climate response to enhanced $CO_2$ in UKESM1. Large continental increases in surface warming

of up to 2 K in the last 50 years of an *abrupt4xCO2* simulation were found due to this effect with the EffCS being 0.3 K higher. The enhanced warming was found to occur predominantly in northern hemisphere mid to high latitudes where the $CO_2$ fertilisation effects lead to the expansion of the (less reflective) boreal forests replacing (more reflective) grasses. As these regions have seasonal snow cover, the surface albedo feedback will likely be influenced by the retuning of the snow burial of vegetation parameter (Andrews et al., 2019). Indeed, spatial plots of the decomposed LW and SW clear-sky and cloudy sky

feedbacks (calculated as the local flux change with global mean surface air temperature change, (Andrews et al., 2019)) indicate a less positive (less amplifying) SW clear-sky feedback across NH continents (Supplementary Figure S9). This is offset by a corresponding increase in the LW clear-sky feedback leading to a negligible impact on the net feedback. Overall the global feedback parameters for the individual components (Figure 18d) show little change from UKESM1 (see Table 2 in Andrews et al. (2019)).

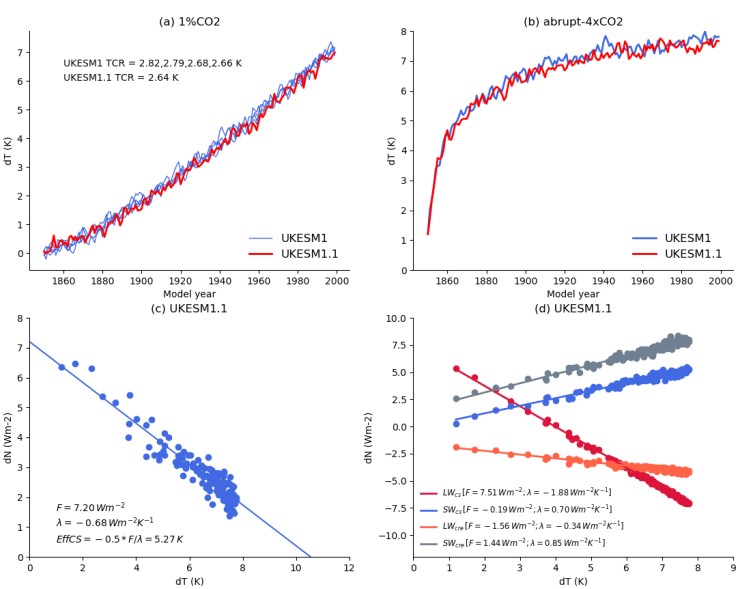

**Figure 18.** Evolution of the global annual mean surface air temperature change in the (a) *1%CO2* and (b) *abrupt4xCO2* simulations from UKESM1 (blue line) and UKESM1.1 (red line). (c) Regression of the change in net downward radiative flux, N, at the top of the atmosphere radiation against the change in surface temperature for UKESM1.1 and (d) as (c) but for the individual SW and LW clear-sky and cloudy sky components.

## 4.4 Future projections

The response of UKESM1.1 to two future emissions scenarios, SSP1-2.6 and SSP3-7.0 are now examined. Figure 19 shows the global mean near surface air temperature anomaly for the historical period (1850 to 2014) and the future projection period (2015 to 2100) from UKESM1 and UKESM1.1 for both SSP1-2.6 and SSP3-7.0. Anomalies are calculated relative to a reference period of 1850-1900 and a five year running mean has been passed through each timeseries before the anomalies are calculated.

The differences in the historical period echo what is discussed in Section 4.2.1 with UKESM1 exhibiting a stronger cooling between 1940 and 1990 and a slightly stronger global rate of warming after 1990. This behaviour continues into the early period of the projections. Beyond 2040 the differences become negligibly small and the temperature timeseries of the two models become very similar for both SSPs. This highlights the smaller role of $SO_2$ emissions in these specific future climate scenarios.

Figure 20 presents maps of the near surface air temperature anomaly calculated as the 2070 to 2100 mean minus the 1850 to 1900 mean, for SSP1-2.6 and SSP3-7.0. Also shown are the difference between the anomalies (UKESM1.1 minus UKESM1). Over most of the globe differences in the air temperature anomalies are less than $\pm 0.25$ K. There is some in-



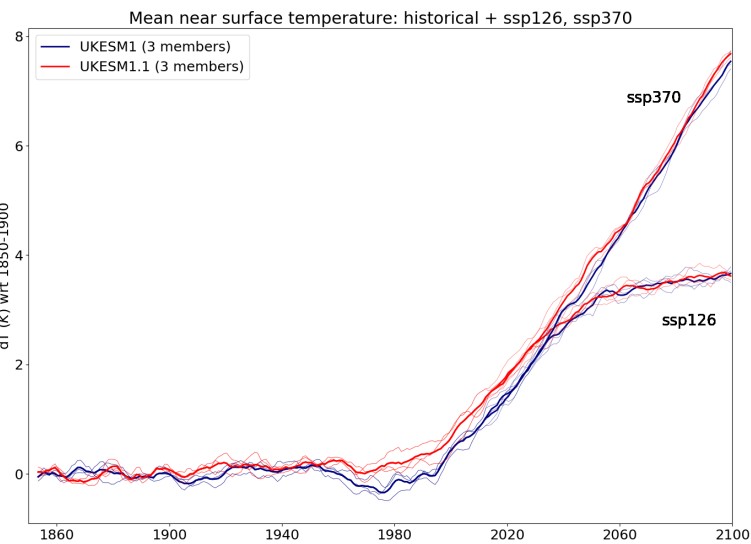

**Figure 19.** Global mean near surface air temperature anomaly relative to 1850-1900 mean for the historical period and SSP1-2.6 and SSP3-7.0 for UKESM1 (blue) and UKESM1.1 (red). Thick lines are the respective three member ensemble means and the thin lines are the individual ensemble members.

dication of UKESM1.1 warming less than UKESM1 over the high northern latitudes, with this tendency reversed over the Arctic ocean and Antarctica. At the beginning of the scenario period UKESM1.1 has thinner sea ice than UKESM1, hence

the fractional sea ice cover lost in UKESM1.1 will be greater than in UKESM1 driving the slightly increased warming over the majority of historical sea ice covered regions (Supplementary Figure S10). The slightly reduced warming over northern hemisphere land regions in UKESM1.1 may reflect differences in the snow albedo feedback associated with our modification to how snow burial of vegetation is treated. We note that differences in the anomalies in high latitude regions are small in comparison to the magnitude of the simulated anomaly itself.

Supplementary Figure S11 shows the zonal mean precipitation anomalies (2070 to 2100 mean) minus (1850 to 1900 mean) for both SSPs and UKESM configurations. Differences are extremely small indicating the two model configurations are largely equivalent in terms of their precipitation sensitivity to future climate change. This is true for both SSPs.

## 5   Discussion and conclusions

This paper describes and evaluates an updated configuration of UKESM1, which we refer to as UKESM1.1. This model repre-

sents a modest update to the UKESM1 model used in CMIP6. One of the primary objectives of this work was to investigate if improvements to the parameterization of $SO_2$ dry deposition, first described in Hardacre et al. (2021), would have a significant





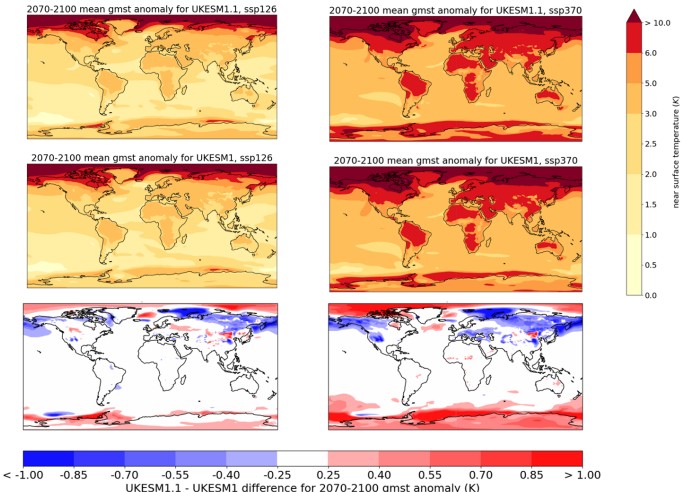

**Figure 20.** Anomalies in near surface air temperature expressed as (2070 to 2100 mean) minus (1850 to 1900 mean) for (left) SSP1-2.6 and (right) SSP3-7.0 simulated by (top) UKESM1 and (middle) UKESM1.1. Bottom row: UKESM1.1-UKESM1 anomaly differences for each SSP.

impact on the large cold bias in thesimulated historical global mean surface temperature in the latter half of the 20th century in UKESM1. Aerosols tend to cool the climate and have likely offset a significant fraction of the warming due to increasing greenhouse gases over the historical era. Given the predominant source of aerosols from $SO_2$ emissions during this time it is

reasonable to expect that any changes to $SO_2$ sources or sinks would impact the anthropogenic aerosol forcing particularly in the late 20th century. While the $SO_2$ dry deposition changes represent the main science change implemented in UKESM1.1 a number of other bugfixes primarily related to aerosols were also included. Furthermore, several tuned parameters in the fully coupled ESM described in Sellar et al. (2019) were revised. The most significant of these include a retuning of the mineral dust scheme to improve the simulation of dust optical depth and a retuning of the extent to which vegetation is buried by snow.

The UKESM1.1 model has a stronger (more positive) total anthropogenic ERF ($1.84\,\mathrm{Wm^{-2}}$ versus $1.76\,\mathrm{Wm^{-2}}$) driven primarily by a weaker (less negative) aerosol ERF ($-1.01\,\mathrm{Wm^{-2}}$ versus $-1.09\,\mathrm{Wm^{-2}}$). The weaker aerosol forcing comes mainly from a weaker clear-sky aerosol scattering contribution. This drives warmer surface temperature anomalies in the northern hemisphere extra-tropics over the historical period and significantly improves the simulation of the historical global mean surface temperature anomaly compared to UKESM1. The weaker aerosol forcing also leads to an increase in ocean heat

uptake in the northern hemisphere oceans along with lower sea ice extent and volume in the Arctic region.

As aerosols are the primary driver of the different climate response in UKESM1.1 we evaluate how the present-day aerosol and related cloud properties compare against observations. Overall, biases and root mean square errors in AOD are improved and the global reduction in cloud droplet number concentration appears to be in better agreement with the satellite data, although there are large observational uncertainties in this derived quantity. More generally, the UKESM1.1 climate evaluates

better or shows neutral changes with respect to UKESM1. Temperature trends since 1900 show an overall improvement al-



though the rate of warming post 1990 is still too high. This is because the climate sensitivity of the model remains largely unchanged. The change in ocean heat content in the top 700 m of the ocean is small but larger changes (increases) are found through the full ocean depth and are in better agreement with a number of OHC datasets. While the reduced Arctic sea ice is also in better agreement with observations and reanalysis datasets, the high climate sensitivity of both models results in too

fast a rate of sea ice loss. Other components of the ESM evaluated remain largely unchanged. For example, minimal changes in ocean circulation are found while some small variations in ocean productivity are evident likely driven by a combination of the warmer climate and increased dust deposition.

Independent tests (not shown) of the impact of the updated $SO_2$ dry deposition scheme only on the aerosol ERF reveal a weakening of the net all-sky aerosol ERF of approximately +0.2 $\mathrm{Wm}^{-2}$ compared to +0.08 $\mathrm{Wm}^{-2}$ found here. In these tests

the weaker aerosol forcing is being driven largely by a weaker aerosol-cloud forcing as opposed to the predominantly clear-sky effects found in this study. A weaker aerosol-cloud forcing is consistent with the smaller positive anomalies in $N_d$ simulated over the historical period in UKESM1.1 but highlights a degree of compensation arising from the other aerosol improvements included in the final configuration. In particular, the important role of the pre-industrial aerosol climate state on anthropogenic aerosol forcing is evident.

The forcing due to land use change remains largely unchanged although a weaker clear-sky ERF is found in response to the retuning of the snow burial of vegetation. The model is clearly sensitive to this highly unconstrained parameter with some regional changes in the climate feedback parameter also found along the marginal snow line. Constraining this parameter using satellite observations of the clear-sky top-of-atmosphere SW radiation reveals a shift from a net positive bias to a net negative bias over the 30° to 60° N latitude region in UKESM1.1. So changes in the future climate response as a consequence of this

retuning is possibly exaggerated here, particularly on the regional scales.

Globally the climate feedback parameter is very similar between the two models, leading to negligible changes in the effective climate sensitivity and transient climate response. Due to the expected smaller role of aerosols in future climate change, the impact of the model updates implemented in UKESM1.1 on future climate assessed through SSP1-2.6 and SSP3-7.0 is found to be small on the global scale. While only a limited subset of the SSPs were assessed here using a relatively small

ensemble of simulations, the results nevertheless provide some assurance of the continued relevance and value of the more extensive set of scenario projections run with the UKESM1 model for CMIP6.

The historical cold temperature bias has affected many CMIP6 models, not only UKESM1. The magnitude of this bias has caused some concern across the community with respect to the fidelity of these complex global climate models and their ability to realistically simulate future climate change. While we do not necessarily support the heavy weighting applied to a

single metric in assessing the skill of an Earth system model, we are interested in understanding the key drivers behind this bias, in particular the role of aerosol. The UKESM1.1 model configuration demonstrates how a seemingly small number of model changes can have a significant impact on the simulated global surface temperature and highlights the large sensitivity of historical climate to aerosols and in particular, to simulated burdens of $SO_2$, a key anthropogenic aerosol precursor. This is consistent with the work of Zhang et al. (2021) who demonstrate a strong correlation between sulphate aerosol loading and the

global mean surface temperature anomalies across a number of the CMIP6 models.





UKESM1.1 offers a notably improved simulation of several key climate variables compared to the UKESM1 model and fixes a number of bugs that were uncovered in UKESM1. While the ECS remains at the upper range of the CMIP6 models, driving too high warming rates in the recent past, the improved simulation of the historical temperature and weaker aerosol forcing will be of interest to many users and scientists.

*Code and data availability. Model code:* All simulations used in this work were performed using version 11.7 of the Met Office Unified Model (UM), version 5.0 of JULES, NEMO version 3.6, CICE version 5.2.1 and OASIS-MCT version 3.0. The full list of simulation identifiers are provided in Table A1 and details of how to access and run the model can be found at https://cms.ncas.ac.uk/unified-model/configurations/ukesm/relnotes-1.1/. NEMO is available to download from http://www.nemo-ocean.eu (last access: March 2022), and the CICE5 model code used here is available from the Met Office code repository at https://code.metoffice.gov.uk/trac/cice/browser (last access:

March 2022). Due to intellectual property right restrictions, we cannot provide the source code for the UM or JULES. The UM is available for use under licence. A number of research organizations and national meteorological services use the UM in collaboration with the Met Office to undertake atmospheric process research, produce forecasts, develop the UM code and build and evaluate Earth system models. To apply for a license for the UM, go to https://www.metoffice.gov.uk/research/approach/modelling-systems/unified-model (last access: March 2022), and for permission to use JULES, go to https://jules.jchmr.org (last access: March 2022).

*Model data:* The UKESM1 simulation data used in this study are archived on the Earth Sytem Grid Federation (ESGF) node (https://esgf-node.llnl.gov/projects/cmip6/). The data citations for UKESM1 and UKESM1.1 are also provided in Table A1.

*Analysis code:* The analysis code and associated model data used to produce the figures in this manuscript is available from Mulcahy (2022).

*Observational data:* EN.4.2.2 data were obtained from https://www.metoffice.gov.uk/hadobs/en4/ and are ©Crown Copyright, Met Office,

2021, provided under a Non-Commercial Government Licence (http://www.nationalarchives.gov.uk/doc/non-commercial-government-licence/version/2/). The HadCRUT5 data was obtained from https://www.metoffice.gov.uk/hadobs/hadcrut5/. Satellite derived AOD data was obtained from ESA CCI (de Leeuw, G and T. Popp and ESA Aerosols CCI project team , 2020; de Leeuw et al., 2020) (last access: 23 June 2020). MODIS AOD data was obtained from https://earthdata.nasa.gov (last access: 9 February 2022). Ground-based AOD data from AERONET are available from https://aeronet.gsfc.nasa.gov/ (last access: 9 February 2022). Cloud droplet number concentration products

are available from http://data.ceda.ac.uk/badc/deposited2018/grosvenor_modis_droplet_conc/ (Grosvenor et al., 2018) and https://doi.org/10.15695/vudata.ees.1 (Bennartz and Rausch, 2017) (last access: 9 February 2022). The RAPID-MOCHA array dataset was obtained from https://www.rapid.ac.uk/data.php (last access: 9 February 2022). Dissolved inorganic nitrogen data was obtained from the World Ocean Atlas 2013 https://www.nodc.noaa.gov/OC5/woa13/ (last access: 9 February 2022). Primary production products (VGPM, Eppley-VGPM, and CbPM) were obtained from the Oregon State University Ocean Productivity group, http://sites.science.oregonstate.edu/ocean.productivity/

(last access: 9 February 2022). Data from the CASTNet (https://www.epa.gov/castnet, last access: 11 February 2022) and EMEP networks (http://ebas.nilu.no/, last access: 11 February 2022) was used in the evaluation of $SO_2$ and $SO_4$ aerosol.



**Table A1.** Summary of the simulation identifiers used in this study. The branch dates off the piControl simulation are shown in parentheses. The SSPs were branched off the first 3 historical simulations listed. Data citations for the model data are also provided.

| Experiment | UKESM1 | UKESM1.1 |
|---|---|---|
| Data citation | Tang et al. (2019); Good et al. (2019) | Mulcahy et al. (2022); Walton et al. (2022) |
| piControl | u-aw310 | u-by230 |
| Historical | u-bc370 (2120-01-01) | u-by791 (2811-01-01) |
| | u-bc292 (2165-01-01) | u-bz502 (2851-01-01) |
| | u-bd483 (2210-01-01) | u-bz897 (2891-01-01) |
| | u-bc179 (2250-01-01) | u-ca306 (2931-01-01) |
| | u-bc470 (2285-01-01) | u-ca811 (2971-01-01) |
| | u-bd288 (2340-01-01) | u-cb799 (3011-01-01) |
| 4xCO2 | u-bb446 (1960-01-01) | u-bz608 (2851-01-01) |
| 1%CO2 | u-bb448 (1960-01-01) | u-bz609 (2851-01-01) |
| | u-bd334 (2120-01-01) | |
| | u-bd335 (2285-01-01) | |
| | u-bd336 (2460-01-01) | |
| SSP1-2.6 | u-be862 | u-cb261 |
| | u-be679 | u-cb584 |
| | u-bu159 | u-cb586 |
| SSP3-7.0 | u-be684 | u-cb180 |
| | u-be690 | u-cb581 |
| | u-bs186 | u-cb585 |

## Appendix A: Simulation and model data information

*Author contributions.* JPM led the development of the UKESM1.1 configuration, the analysis and wrote the manuscript. CGJ initiated the UKESM1.1 project, led the development of the $SO_2$ dry deposition scheme and made significant contributions to the development and analysis of UKESM1.1. STR coded up the new dry deposition scheme and ran numerous model experiments. TK, AJD, EWB, AY carried out analysis and made significant contributions to the text. JW, CH, TA, ABS contributed figures and/or analysis included in this manuscript. MS, LdM, PH, RH, DK, ER and YT made invaluable contributions to the development of the UKESM1.1 configuration. All co-authors contributed comments on this manuscript.


*Competing interests.* The authors declare they have no competing interests





*Acknowledgements.*   JPM, JW, EWB, TA, CH, YT, RH, ABS and ER were supported by the Met Office Hadley Centre Climate Programme
funded by BEIS. CJ, SR, TK, DK, PH, LDM, AY were funded by the National Environmental Research Council (NERC) national capa-
bility grant for the UK Earth System Modelling project, grant NE/N017951/1, and were also supported by the EU Horizon 2020 Research
Programme CRESCENDO project, grant agreement 641816. A.J.D. was funded by the UK NERC REAL project (NE/N018591/1)

We thank Matthew Palmer and Rachel Killick for their support in obtaining the observational data sets for ocean heat content. We thank
Daniel Grosvenor and Adam Povey for help obtaining the cloud droplet number concentration and satellite aerosol optical depth datasets
respectively.





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
