# Peer review of "UKESM1.1: Development and evaluation of an updated configuration of the UK Earth System Model"

_Geoscientific Model Development, 2022_

## Referee Comment (RC2)

This is an important manuscript that describes the changes made between UKESM1 and UKESM1.1 in order to improve the simulation of the historical surface temperature in the second half of the 20[th] century. A number of changes and bug fixes were made, but the key change appears to be a reduction in the magnitude of the aerosol ERF as a result of a reduction is sulphate.

The discussion focuses on one specific model but the problem of an overly cold late 20[th] century is present in other climate models as well. Therefore the manuscript should be relevant to a broad audience and well suited for GMD.

While I believe that the conclusions are very likely correct, thie analysis does not currently provide sufficient evidence to support them. The reported change in ERF is small difficult to attribute to aerosol only (see major comment below).

I recommend performing additional simulations (time evolving ERF calculations) to better support the conclusion that the primary reason for the improvement in surface temperature is due to a reduction in the aerosol forcing.

**Major comment**

The magnitude of the change in total ERF (+0.08 W/m2) appears small compared to the actual change in surface temperature (Fig 3a).

Using a simple back-of-the-envelope calculation (see Shindell 2014, doi:10.1038/nclimate2136), we can estimate the warming for a given forcing and TCR as:

dT = TCR/F2xCO2 ERF_tot

where TCR is the transient climate response, F2xCO2 the 2xCO2 forcing, ERF_tot the total anthropogenic forcing. Let's assume  F2xCO2 = 3.6 W/m2 for both UKESM1 and 1.1 (based on Figure 18c) and estimate dT for both models:

dT_UKESM1 = 2.76 / 3.60 * 1.76 = 1.35 K
dt_UKESM1.1 = 2.64 / 3.60 * 1.84 = 1.35 K

Based on that simple calculation, both models would yield about the same level of warming due to a compensation between an increase in total forcing and a reduction in TCR. While that is not the case, it does make it difficult to simply conclude the all the changes arise from ERF while the TCR remains "essentially unchanged" (line 14).

Another way to look at this to calculate how large a temperature change one might expect given the change in ERF (0.08 W/m2):

dT = 2.64 / 3.60 * 0.08 = 0.06 K

This value is very small compared the actual temperature difference between the models (Fig 3a).

The most likely explanation for this discrepancy is that the difference in ERF is much larger during the period 1960-1990 than the value of 0.08 W/m2 reported for 2014.

Similarly, the forcing values presented in Table 3 are not very convincing. The total anhtropogenic forcing is indeed larger in UKESM1.1 (+1.84 W/m2) than UKESM1 (+1.76 W/m2). However, summing the components yields a smaller forcing for UKESM1.1 (+1.61 W/m2) than UKESM1 (+1.65 W/m2), and both of them are off by more than the difference between models. I don't think this data supports the conclusion that the change is aerosol forcing is key. Having comparable values for the period 1960-1990 would likely help.

I would recommend to perform additional simulations to estimate ERF for different periods more relevant to the cold bias. The best would be to follow RFMIP experiments for diagnosing time-evolving ERF. Due to the need for additional simulations, I recommend that the manuscript be returned for major revisions.

**Minor comments**

Lines 122-127: paragraph requires clarification. If I understand correctly, $r_c$ was set to 10 sm-1 in GC3.1, then mistakenly to 148.9 sm-1 in UKESM1 and then to 1 sm-1 in UKESM1.1. Clarify the motivation for using 1 sm-1 instead of 10 as in GC3.1? Insufficient $SO_2$ dry deposition?

Lines 207-211: What is the impact on net TOA radiation?

Line 271: any reason for stopping at 462 years and not the recommended 500 years for DECK piControl?

Lines 275-276: "later period". Chosen because of the smaller drift or for other reason?

Figure 3: HadCRUT5 reports SST over ice-free ocean, and surface air temperature over land and ice covered ocean. Was the same calculation done for the model output?

Line 319: 1900 → 1901 for consistency with the figures. Similar on line 324.

Lines 355-356: explain how globally averaged Nd and $r_{eff}$ were calculated.

Figure 6: how different are the starting values?

Lines 365-372: are Nd anomalies really relevant if the clear-sky OSW anomalies are driving the surface temperature change?

Figure 9: explain how vertically averaged Nd was calculated.

Line 418: is the detrending actually needed? piControl looks stable in Figure 1b.

Line 425: "relatively large climate sensitivity" here, and "outside of the CMIP6 5-95% ranges" on line 552.

Table 4: use a consistent number of decimals and verify that the net adds up, or explain why.

Line 602: "thesimulated" → "the simulated"

Lines 610-611: Table 3 currently doesn't support this assertion. See major comment above.

---

## Author Comment (AC3)

**Response to Review by RC2**

**Relevant figures to accompany this response:**

[Figure]

*Figure: (left) Timeseries of the aerosol ERF from UKESM1 (blue) and UKESM1.1 (red) for all-sky (solid) and clear-sky (dashed lines) conditions. The aerosol ERFs are calculated for selected timeslices along the historical period; (right) the interhemispheric gradient in aerosol ERF for each timeslice.*

**Comparison of using tas versus blended tas/sst dataset for figure 3:**

[Figure]

Figure (above): Surface air temperature anomalies using only 'tas'.

[Figure]

Figure (above): Surface air temperature anomalies using a blended 'tas' and SST dataset.

---

## Author Response (AR1)

**Response to Review by RC1, Dr. Andrew Gettleman.**

We thank Dr. Gettleman for taking the time to review our manuscript and for his constructive comments on our paper. We respond to each point below, where the reviewer comments are in **black** and our responses are in **blue.** Line numbers (denoted by **LXXX** ) refer to Line Numbers in the revised manuscript accompanying this resubmission.

**RC = Reviewer Comment**
**AR = Authors Response**

**RC:** This manuscript is a comprehensive assessment of updates to the UKESM1, documenting a new version of a major Earth System Model. The manuscript is well written and should be acceptable for publication in Geoscientific Model Development with minor revisions. One general comment is that it's still unclear to me how you are ascribing sensitivity of the model to SO2 and SO4 partitioning, and the processes that are acting. You talk a lot about deposition, but very little about oxidation rates. See detailed comments below. This could be better clarified in the manuscript. Otherwise this is an excellent and comprehensive treatment of model evaluation.

**AR:** Thank you for your positive comments on our manuscript. We address your specific comment on SO2 and SO4 partitioning in your specific comments below.

**Specific comments are below.**

**RC: Page 1, L12: note this is a reduction in magnitude of aerosol ERF (which is negative)**

**AR:** We have reworded this sentence as follows:
**L11:** *"Changes to the aerosol and related cloud properties are a driver of the improved GMST simulation despite only a modest reduction in the magnitude of the negative aerosol ERF (which increases by +0.08 Wm-2)."*

**RC: Page 3, L59-63: This discussion is confusing. I read it 3 times and it still doesn't really make sense. Is the problem SO2 deposition or SO2 oxidation? It seems to be both, but you just say it's SO2 deposition I think. Please clarify.**

**AR:** Thank you for highlighting the lack of clarity here. It is indeed the point that both deposition and oxidation processes are uncertain in the model. We hypothesise in the previous paragraph that too little removal of SO2 (via potentially both deposition and oxidation processes) close to source leads to excess transport of SO2 to remote regions where it is eventually oxidised to SO4, leading to a potential positive bias in remote SO4 aerosol. Given the remote marine location this has a large effect on the aerosol forcing. This work implements updates to the dry deposition only and we hope to investigate oxidation processes in the future, so the intention was just to highlight oxidation processes as another source of uncertainty here.

We have reworded this paragraph to make this more clear.

**L55-64: "***Hardacre et al (2021) examine the impact of an updated parameterization for the dry deposition of SO2 on the surface SO2 concentration bias in UKESM1. The new parameterization considers whether the surface vegetation is wet or dry when calculating the surface resistance to species uptake. Due to the high solubility of SO2, the wetter and more humid at the surface the higher the uptake of SO2. The new parameterization leads to a significant improvement (of the order*

*of 50%) in the positive SO2 bias against ground-based observations in the above study. Despite this improvement in the simulation of surface SO2, the reductions of SO2 close to source further degrade the pre-existing low bias in SO4 aerosol (Hardacre et al. 2021, Mulcahy et al. 2020) and so model process deficiencies in the oxidation of SO2 to SO4 also likely exist.*
*Interestingly, Hardacre et al. (2021) show a larger relative reduction in surface SO2 and SO4 remote from source (e.g. over the North Atlantic region) than over the source regions supporting the above assertion that excess SO2 close to source regions drives remote aerosol loading and subsequent aerosol forcing."*

**RC: Page 3, L70: where is section 3?**

**AR:** Many thanks for highlighting this omission. Section 3 is now referenced in this paragraph.

**L70:** "*….Section 3 details the model simulations conducted as part of this study.*"

**RC: Page 3, L80: So is GA 7.1 also part of HadGEM3-GC3.1? The terminology is a bit confusing.**

**AR:** Yes, GA7.1 is the Global Atmosphere component of GC3.1 and UKESM1. We apologise for the confusion. We have modified the text to clarify that both models have largely the same atmosphere:

**L82:** "*The physical atmosphere component (including aerosol) of UKESM1 (and GC3.1) is the Global Atmosphere 7.1 science configuration of the Unified Model …..*"

**RC: Page 5, Table 1: How is DMSO + OH —> 0.6SO2 a balanced chemical reaction for S? Also, might note you have neglected things in these reactions (eg. DMS + OH —> SO2). Maybe better to have the whole reactions here?**

**AR:** Thank you for your comment. Technically speaking the reaction should be DMSO+OH ->0.6SO2+0.4MSA (Pham et al. 1995). MSA is an inert tracer in UKESM1 and represents an effective sulfur sink. You are correct that the DMS chemistry (even with this new change) in the model is a gross over-simplification of the actual (highly complex) DMS chemistry taking place in the atmosphere. Many intermediary products are neglected (e.g. the potentially important role of halogens). This change seeks to make the DMS chemistry consistent with the physical model and reintroduces the important DMSO intermediary (Chen et al.,2018, Pham et al. 1995), which is a tracer in the UKCA-StratTrop model anyway and undergoes wet and dry deposition. We have corrected the reaction in Table 1 of the updated manuscript and highlighted the general simplicity of the scheme in the text:

**L136:** "*Currently, in the gas phase, DMS is oxidised by OH via an abstraction and addition pathway. The addition pathway, favoured at lower temperatures, neglects the formation of the intermediary product, dimethyl sulfoxide (DMSO), despite this being a transported tracer which undergoes wet and dry deposition in the StratTrop scheme. In UKESM1.1, the DMS chemistry is updated to include the formation of DMSO as shown in Table 1 and is now consistent with GC3.1. The DMS chemistry in UKESM1.1 remains a highly simplified scheme, Revell et al. (2019) investigated the impacts of more complex DMS chemistry on SO4 aerosol production and found a notable impact on cloud droplet number concentrations in the Southern Ocean.*"

**RC: Page 5, L141: does the SO4 go into a different mode?**

**AR:** The bug led to sulphuric acid tendencies from the chemistry scheme not being updated on the shorter microphysical timesteps controlling nucleation, condensation and coagulation processes. This resulted in too high an initial H2SO4 concentration at the start of the chemistry timestep and subsequent excessive number concentration of small particles being produced by the nucleation process. The bugfix distributes this concentration correctly across the smaller substeps reducing the amount of H2SO4 initially available for nucleation, so nucleation mode number concentration decreases while there is a small increase in the condensation sink onto other modes. The bugfix and impacts of fix are documented in Ranjithkumar et al. 2021.

**RC: Page 6, L152: maybe add a sentence on how these values were derived from AMIP runs? What was the methodology in brief?**

**AR:** We have now added the following:

**L158:** "*Here, numerous AMIP simulations were conducted with the parameters of interest independently adjusted and outputs evaluated against observations. *"

**RC: Page 7, L164: 'snow metric' is strange. Just call it the TOA outgoing clearsky SW flux over land…**

**AR:** The terminology is consistent with what was used in Sellar et al. 2019, but we are happy to change it here if it is clearer:

**L170:** "*When evaluated in present-day simulations however, this tuning appears to lead to an excessive burial by snow and results in a net positive bias in the DJF top-of-atmosphere clear-sky outgoing SW radiation between 30N and 60N (see Table 2 Sellar et al. 2019).*"

**RC: Page 7, L172: so the tuning darkens the present so it is warmer and does not change as much in the future?**

**AR:** Yes, the positive albedo feedback is smaller.

**RC: Page 7, L174: Capitalize Dust Optical Depth**

**AR:** Now corrected (L182)

**RC: Page 8, L199: What does it mean that 'tuning was omitted'? A parameter value came from somewhere. Is it that the protocol suggested that sub-grid gravity wave flux be adjusted to get the right period of the QBO? Or was this found after UKESM1 was released? Please be a bit more descriptive of the process.**

**AR:** The USSP_launch_factor represents the generation of vertically propagating gravity waves by tropospheric convection and is sensitive to both model resolution and science configuration. It generally requires retuning for a change in model resolution and / or new science developments. By "omitted" we mean that when developing the N96ORCA1 configuration of GC3.1 (and subsequently UKESM1.0 at the same resolution) we neglected to tune this parameter and inherited the value from the higher-resolution N216ORC025 model.  We have changed the text to make this clear:

**L206:** *The parameter (USSP_launch_factor) controlling the flux of sub-grid gravity waves generated by non-orographic sources is sensitive to both model resolution and science configuration and generally requires retuning when changing model resolution or implementing new science (Walters et al. 2014). This retuning was erroneously neglected during the development of UKESM1 which*

*subsequently inherited the value of USSP_launch_factor used in the higher resolution physical model. As a consequence the period of the tropical quasi-biennial oscillation (QBO) was found to be too low in UKESM1 when compared against reanalyses (Richter et al., 2020)."*

**RC: Page 8, L204: mean QBO period…..(also line 205)**

**AR:** Now corrected (L214/215).

**RC: Page 8, L211: What does the parameter do? I guess the tuning controls the LWP and the SW cloud radiative effect?**

**AR:** The parameter, two_d_fsd_factor, describes the assumed sub-grid scale cloud water inhomogeneity. Larger values assume a greater sub-grid inhomogeneity of grid box mean cloud water and thus a less reflective cloud (vice versa for decreased values of this parameter). Development of the *two_d_fsd_factor* parameter is detailed in Hill et al. (2015). Due to this parameter being extremely poorly constrained by observations and its impact on TOA SW fluxes it is often used as a final tuning term for achieving a balanced TOA budget. We have amended the text as follows:

**L221:** *"Increasing the parameter value translates to a greater assumed sub-grid inhomogeneity of grid box mean cloud water and thus a less reflective cloud (and vice versa for decreased values of this parameter) although only a small retuning of this parameter - from a value of 1.48 to 1.49 - was required here."*

**RC: Page 9, L229: what are the SSTs in the piClim-control**

**AR:** SSTs and sea-ice in the piClim-control are taken from a 30 year period of the fully coupled piControl simulation (and represent the mean over the 30 year period). We have clarified this in the text:

**L243:** *"This configuration follows the Radiative Forcing Model Intercomparison Project (RFMIP, Pincus et al. 2016) protocol and takes simulated SST, sea-ice fields as well as the other climatological forcing fields described above from the piControl simulation. All other prescribed forcing data is also from 1850"*

**RC: Page 9, L234: what is the second piClim-control-2014 experiment called?**

**AR:** This experiment is called *piClim-anthro* – this has now been clarified in the text (**L246**).

**RC: Page 20, L401: what is 'mean Nd' averaged over within the column? It's given as a concentration per unit mass, so it's not column. Averaged over cloud layers?**

**AR:** For this analysis we diagnose the Nd at cloud top to enable a more accurate comparison with satellite retrievals. We have clarified this in the text , see **L367/368 and L425, Figure 9 Caption** while the Caption of **Figure 6** now includes the following sentence:
*"Both cloud droplet number concentration and effective radius represent cloud-top values"*

**RC: Page 21, L435: Does the improvement when below 700m is included indicate that OHC is not partitioned at the right layers in UKESM?**

**AR:** Yes, in the historical simulations the vertical distribution of the additional heat in the ocean is not in full agreement with observations. In the period after 1991, there is too much heat stored in the 0-700 m layer, and not enough in the layers deeper than 700m. A much more detailed analysis of the historical ocean heat content changes in UKESM1.0 is given in Kuhlbrodt et al. (2022). We have adjusted the sentence to be more clear:

**L447:** *"This implies that in the ocean layers below 700m the uptake of heat is too small during this period, compensating for the overly strong increase above 700m."*

**RC: Page 23, L465: what is the mechanism by which weaker aerosol forcing lowers AMOC? That does not seem trivial or obvious. Please explain how this is 'consistent'**

**AR:** The mechanisms were proposed by Menary et al. (2013) based on a study using HadGEM2-ES, the predecessor model of UKESM1. Aerosol forcing induces atmospheric circulation changes over the North Atlantic and Arctic, which ultimately lead to an increase in the salinity in the North Atlantic, which decreases the stability of the water column, driving an increase in the overturning circulation. The increase in salinity seems to come from a combination of a decrease in ice transport through the Fram Straits, increased evaporation over the subpolar gyre and a positive ocean circulation feedback (stronger overturning brings more saline water northwards from the subtropics). Menary et al 2020 (see also Robson et al. 2020) have subsequently attributed the strengthening of the AMOC over the historical period (1850-1980) to the magnitude of the aerosol forcing across several CMIP6 models. They developed a metric for aerosol forcing of the AMOC that is proportional to the hemispheric gradient of net downward SW at TOA. With increased anthropogenic aerosol loading the northern hemisphere becomes increasingly more reflective than SHEM so net downward SW at TOA in the NHEM decreases while the SHEM stays largely unaffected. This energy imbalance seems (in models) to be balanced by a shift in the ITCZ or by a change in AMOC strength (Marshall et al. 2014). The strength of the AMOC is systematically weaker in UKESM1.1 (by a small amount, <10% ) and this is consistent with a less negative (weaker) aerosol forcing. For completeness we have added the Menary et al. (2013) and Robson et al. (2020) references to the revised manuscript (see **L476**) and have also changed the language from "a weaker aerosol forcing" to "a less negative aerosol forcing" on **L478**.

**RC: Page 25, L496: For the Antarctic sea ice you state there is no significant difference between UKESM1 and UKESM1.1. But is there an increase or decrease over time, or no change? And how does that compare to observations.**

**AR:** Thank you for your comment. We have now included additional detail on the model performance in the Antarctic:

**L510:** *"Both models simulate a flat trend in both extent and volume up until the late 1970s after which the extent and volume decrease at similar rates in both models. Observations of sea ice extent from 1979 show a small positive trend which is not captured by the models."*

**RC: Page 27, L527: does 'stronger' mean less negative? If so, awkward. It's actually a reduction in magnitude. Please clarify. Also the 'increases' is a decrease in magnitude right? (Less negative). Might be more clear to use magnitude.**

AR:  We apologise for the confusion in terminology. By 'stronger' we mean more negative and by 'weaker' we mean a less negative aerosol forcing. We agree this is confusing and so have adjusted the language to refer to "a less negative ERF" or "more positive" or "a more negative ERF" etc in this section (see **L538, 541, 542, 561, 562)**

**RC: Page 27, L529: why is the aerosol effect positive over China and India?**

**AR:** This has been documented in O'Connor et al (2021) and is due to the strong absorption by BC in UKESM1 resulting in regional positive forcings. O'Connor et al. (2021) show how this comes through the instantaneous radiative forcing as the SW and LW cloud adjustments were found to cancel (see also Johnson et al. 2019).

**RC: Page 35, L658: do you want to comment on TCR being high as well? Also you might note that the cold temperature bias does not seem to be related to high ECS, since changing it did not alter ECS.**

**AR:** Thank you for this suggestion. We have adjusted this sentence to state :
**L684:** "*While the ECS and TCR remain at the upper range of CMIP6 models….*"
We have also added the following sentence to the Discussion section of the similarity between feedback parameters of the two model at:
**L666**: "*Furthermore, the similarity in the effective climate sensitivity and transient climate response demonstrates that, in this model at least, the effective climate sensitivity does not seem to be related to the magnitude of the aerosol effective radiative forcing or the magnitude of the historical cold temperature bias.*"

**References:**

Chen, Q., Sherwen, T., Evans, M., and Alexander, B.: DMS oxidation and sulfur aerosol formation in the marine troposphere: a focus on reactive halogen and multiphase chemistry, Atmos. Chem. Phys., 18, 13617–13637, https://doi.org/10.5194/acp-18-13617-2018, 2018.

Pham, M., Müller, J.-F., Brasseur, G. P., Granier, C., and Mégie, G.: A three-dimensional study of the tropospheric sulfur cycle, J. Geophys. Res., 100, 26061–26092, https://doi.org/10.1029/95jd02095, 1995.

Johnson, B. T., Haywood, J. M., and Hawcroft, M. K.: Are changes in atmospheric circulation important for black carbon aerosol impacts on clouds, precipitation, and radiation? J. Geophys. Res.-Atmos., 124, 7930–7950, https://doi.org/10.1029/2019JD030568, 2019.

Marshall, J., Donohoe, A., Ferreira, D., & McGee, D. (2014). The ocean's role in setting the mean position of the Inter-Tropical Convergence Zone. *Climate Dynamics*, **42**(7), 1967–1979. https://doi.org/10.1007/s00382-013-1767-z

Menary, M. B., Roberts, C. D., Palmer, M. D., Halloran, P. R., Jackson, L., Wood, R. A., Müller, W. A., Matei, D., & Lee, S.-K. (2013). Mechanisms of aerosol-forced AMOC variability in a state of the art climate model. *Journal of Geophysical Research: Oceans*, **118**, 2087–2096. https://doi.org/10.1002/jgrc.20178

Menary, M. B., Robson, J., Allan, R. P., Booth, B. B. B., Cassou, C., & Gastineau, G., et al. (2020). Aerosol-forced AMOC changes in CMIP6 historical simulations. *Geophysical Research Letters*, 47, e2020GL088166. https://doi.org/10.1029/2020GL088166

O'Connor, F. M., Abraham, N. L., Dalvi, M., Folberth, G. A., Griffiths, P. T., Hardacre, C., Johnson, B. T., Kahana, R., Keeble, J., Kim, B., Morgenstern, O., Mulcahy, J. P., Richardson, M., Robertson, E., Seo, J., Shim, S., Teixeira, J. C., Turnock, S. T., Williams, J., Wiltshire, A. J., Woodward, S., and Zeng, G.:

Assessment of pre-industrial to present-day anthropogenic climate forcing in UKESM1, Atmos. Chem. Phys., 21, 1211–1243, https://doi.org/10.5194/acp-21-1211-2021, 2021.

Ranjithkumar, A., Gordon, H., Williamson, C., Rollins, A., Pringle, K., Kupc, A., Abraham, N. L., Brock, C., and Carslaw, K.: Constraints on global aerosol number concentration, $SO_2$ and condensation sink in UKESM1 using ATom measurements, Atmos. Chem. Phys., 21, 4979–5014, https://doi.org/10.5194/acp-21-4979-2021, 2021

Revell, L. E., Kremser, S., Hartery, S., Harvey, M., Mulcahy, J. P., Williams, J., Morgenstern, O., McDonald, A. J., Varma, V., Bird, L., and Schuddeboom, A.: The sensitivity of Southern Ocean aerosols and cloud microphysics to sea spray and sulfate aerosol production in the HadGEM3-GA7.1 chemistry–climate model, Atmos. Chem. Phys., 19, 15447–15466, https://doi.org/10.5194/acp-19-15447-2019, 2019.

Robson, J., Aksenov, Y., Bracegirdle, T. J., Dimdore-Miles, O., Griffiths, P. T., & Grosvenor, D. P., et al. (2020). The evaluation of the North Atlantic climate system in UKESM1 historical simulations for CMIP6. *Journal of Advances in Modeling Earth Systems*, 12, e2020MS002126. https://doi.org/10.1029/2020MS002126

Kuhlbrodt, T., Voldoire, A., Palmer, M. D., Geoffroy, O., and Killick, R. E. (2022). Historical Ocean Heat Uptake in Two Pairs of CMIP6 Models: Global and Regional Perspectives. *Journal of Climate*, https://doi.org/10.1175/JCLI-D-22-0468.1

**Response to Review by RC2**

We thank Reviewer RC2 for their helpful and constructive comments on our manuscript. We respond to each point below, where the reviewer comments are in **black** and our responses are in **blue.** Line numbers (denoted by **LXXX** ) refer to Line Numbers in the revised manuscript accompanying this resubmission.

**RC = Reviewer Comment**
**AR = Authors Response**

**RC:** This is an important manuscript that describes the changes made between UKESM1 and UKESM1.1 in order to improve the simulation of the historical surface temperature in the second half of the 20th century. A number of changes and bug fixes were made, but the key change appears to be a reduction in the magnitude of the aerosol ERF as a result of a reduction is sulphate.

The discussion focuses on one specific model but the problem of an overly cold late 20th century is present in other climate models as well. Therefore the manuscript should be relevant to a broad audience and well suited for GMD.

While I believe that the conclusions are very likely correct, thie analysis does not currently provide sufficient evidence to support them. The reported change in ERF is small difficult to attribute to aerosol only (see major comment below).

I recommend performing additional simulations (time evolving ERF calculations) to better support the conclusion that the primary reason for the improvement in surface temperature is due to a reduction in the aerosol forcing.

**AR:** We thank the reviewer for making some highly relevant points in their review and hope we have satisfactorily address the Major Comments below.

**RC: Major comment**

The magnitude of the change in total ERF (+0.08 W/m2) appears small compared to the actual change in surface temperature (Fig 3a).

Using a simple back-of-the-envelope calculation (see Shindell 2014, doi:10.1038/nclimate2136), we can estimate the warming for a given forcing and TCR as:

dT = TCR/F2xCO2 ERF_tot

where TCR is the transient climate response, F2xCO2 the 2xCO2 forcing, ERF_tot the total anthropogenic forcing. Let's assume F2xCO2 = 3.6 W/m2 for both UKESM1 and 1.1 (based on Figure 18c) and estimate dT for both models:

dT_UKESM1 = 2.76 / 3.60 * 1.76 = 1.35 K
dt_UKESM1.1 = 2.64 / 3.60 * 1.84 = 1.35 K

Based on that simple calculation, both models would yield about the same level of warming due to a compensation between an increase in total forcing and a reduction in TCR. While that is not the case, it does make it difficult to simply conclude the all the changes arise from ERF while the TCR remains "essentially unchanged" (line 14).

Another way to look at this to calculate how large a temperature change one might expect given the change in ERF (0.08 W/m2):

dT = 2.64 / 3.60 * 0.08 = 0.06 K

This value is very small compared the actual temperature difference between the models (Fig 3a).

The most likely explanation for this discrepancy is that the difference in ERF is much larger during the period 1960-1990 than the value of 0.08 W/m2 reported for 2014.

Similarly, the forcing values presented in Table 3 are not very convincing. The total anthropogenic forcing is indeed larger in UKESM1.1 (+1.84 W/m2) than UKESM1 (+1.76 W/m2). However, summing the components yields a smaller forcing for UKESM1.1 (+1.61 W/m2) than UKESM1 (+1.65 W/m2), and both of them are off by more than the difference between models. I don't think this data supports the conclusion that the change is aerosol forcing is key. Having comparable values for the period 1960-1990 would likely help.

I would recommend to perform additional simulations to estimate ERF for different periods more relevant to the cold bias. The best would be to follow RFMIP experiments for diagnosing time-evolving ERF. Due to the need for additional simulations, I recommend that the manuscript be returned for major revisions.

**Author Response (AR):** We thank the reviewer for their comments and agree that the change in the aerosol ERF in UKESM1.1 relative to UKESM1 appears modest (at +0.08Wm-2) and the reviewer rightly questions how this small net change in the aerosol ERF can be the main driver of the improved historical surface temperature evolution in UKESM1.1. Following the reviewer's recommendation, we have now conducted a series of additional aerosol ERF timeslice experiments at different time points during the historical period where we perturb the anthropogenic aerosol emissions to 1900, 1920, 1950 and 1980 conditions respectively, in addition to the 2014 timeslice already conducted. Each timeslice is run for 45 years with the aerosol ERF calculated for the final 30 years as outlined in the manuscript. The aerosol ERF timeseries (for net all-sky but also clear-sky fluxes at the top-of-atmosphere) is shown below (and is included in the manuscript as new Figure 21) and has also been now added to the main text. It shows that post 1920 when anthropogenic SO2 emissions start to rapidly increase the aerosol ERF in UKESM1.1 is consistently less negative in magnitude, with the *change* in aerosol ERF increasing from +0.01Wm-2 in 1920 to a maximum difference of +0.2Wm-2 in 1980. Hence, we agree with the reviewer that the difference in aerosol ERF is bigger between the models at this crucial time-period than was indicated by simply considering the change in ERF between pre-industrial and a 2014 timeslice. We thank the reviewer for the suggestion to investigate this further.

The change in aerosol ERF is predominantly coming from the Northern Hemisphere which shows larger changes in aerosol ERF (changing by +0.1Wm-2 in 1920 to +0.31Wm-2 in 1980) than the Southern Hemisphere. In addition to the aerosol ERF timeseries figure we now also include a plot of the interhemispheric difference in the aerosol ERF for each timeslice (new Fig 21b and shown also below). The smaller interhemispheric gradient in the forcing for all years assessed points to differences in the spatial inhomogeneity of the aerosol forcing in UKESM1.1 which could lead to changes in the transient sensitivity of the aerosol forcing. The role of inhomogeneous forcing is highlighted as important and incorporated in Shindell et al (2014) through their ratio, E, where the calculation of dT above becomes dT=(TCR/F_2xCO2) x (Fghg+E*(Faer+Fo3+Flu)).

[Figure]

*Figure: (left) Timeseries of the aerosol ERF from UKESM1 (blue) and UKESM1.1 (red) for all-sky (solid) and clear-sky (dashed lines) conditions. The aerosol ERFs are calculated for selected timeslices along the historical period; (right) the interhemispheric gradient in aerosol ERF for each timeslice.*

In addition, we believe the different PI background states of UKESM1 and UKESM1.1 is of importance for both the aerosol ERF and also the historical temperature response to forcing. The UKESM1.1 PI climate in the *piClim-Control* simulation is more positive than UKESM1 by +0.56Wm-2. The change is driven by a combination of the large reduction in the outgoing SW (-1.57Wm-2) being offset by an increase in OLR (+0.97 Wm-2). Natural marine sources of DMS dominate the PI distributions of SO2 and subsequent sulphate aerosol. Therefore, changes to the DMS chemistry, SO2 dry deposition parameterization and the bugfixes to the H2SO4 updating and vertical profile of cloud droplet number concentration calculation all alter the PI background state. These changes result in a notable less negative SW cloud forcing in UKESM1.1 PI climate (-43.80 Wm-2 versus -45.02Wm-2). This change in PI climate translates to the fully coupled historical simulations driving a more positive net TOA radiation and subsequent warmer temperatures. We have additionally now included the historical timeseries of the absolute values of the net TOA radiation and its components as well as the SW cloud forcing and cloud fraction to highlight the different base climates more clearly in these two configurations.

In summary, we believe the dT calculated by the reviewer above does not take into account the spatial inhomogeneity of the aerosol forcing and how that may change between UKESM1 and UKESM1.1. Also, it is calculated for 2014 relative to 1850, while the period where the largest change in dT occurs (as the reviewer points out) is closer to 1980 minus 1900. During this period, the anthropogenic aerosol forcing will play a relatively larger role than the homogeneous forcing from GHGs and so will have a stronger relative impact on surface temperature. We hypothesize that the sustained less negative aerosol forcing over the historical period imposed on a warmer PI background climate is an important factor in the improved simulation of historical surface temperature in UKESM1.1. We thank the reviewer for prompting us to carry out this additional analysis which improves the paper and our conclusions. A full transient aerosol ERF simulation is planned for future work and for this manuscript we have modified the text and conclusions in the manuscript to reflect this additional analysis **as follows:**

**L535 (start of Sect 4.3):** *"As noted earlier, the warmer climate in UKESM1.1 and improved representation of the historical surface temperature is believed to be largely due to a weaker anthropogenic aerosol forcing driven predominantly by lower sulphur dioxide and hence sulphate*

*aerosol burdens in the updated configuration"* is changed to : *" We now compare the key effective radiative forcings between UKESM1.1 and UKESM1 and examine their potential role in the improved simulation of historical surface temperature."*

**L537 now states: "***The total anthropogenic ERF is more positive in UKESM1.1 increasing from 1.76Wm-2 to 1.84Wm-2. This is in part due to a less negative aerosol ERF……"*

We have made significant changes to the Discussion and Conclusions section with the main change being the inclusion of a new paragraph and a new figure (Figure 21) of the aerosol ERF timeseries and interhemispheric difference in the aerosol ERF.

**L648:** *"The relatively modest change in global-mean aerosol ERF between UKESM1 and UKESM1.1, calculated for 2014 anthropogenic aerosol conditions relative to pre-industrial, is unlikely to be the sole explanation of the improved historical temperature response. To get a better indication of how the aerosol ERF changed throughout the historical period we have conducted additional aerosol ERF simulations (piClim-aer) for 1900, 1920, 1950 and 1980, in addition to the 2014 simulation. The resulting aerosol ERF timeseries (Figure 21) shows that post 1920 when anthropogenic SO2 emissions start to rapidly increase the aerosol ERF in UKESM1.1 is consistently less negative in magnitude than UKESM1, with the change in aerosol ERF increasing from +0.01Wm-2 in 1920 to a maximum difference of +0.2Wm-2 in 1980. This change in aerosol ERF is predominantly coming from the Northern Hemisphere which shows larger changes in aerosol ERF (changing by +0.1Wm-2 in 1920 to +0.31Wm-2 in 1980) than the Southern Hemisphere leading to a weaker interhemispheric gradient in the aerosol ERF in UKESM1.1 (Figure 21). This change in the regional pattern of aerosol ERF between UKESM1 and UKESM1.1 could imply that the transient sensitivity to aerosol forcing has changed in UKESM1.1 (Shindell, 2014). The dependence of the transient sensitivity on the forcing is often described as an efficacy (Hansen et al., 2005), and in CMIP6 models the radiative feedbacks in response to aerosol forcing have been found to more amplifying (transient sensitivity higher) than that to greenhouse gas forcing (Salvi et al., 2022). We hypothesize that the less negative aerosol forcing over the historical period imposed on a warmer background climate state, which has a less negative SW cloud forcing, is an important factor in the improved simulation of historical surface temperature in UKESM1.1. Our comparison of the effective climate sensitivity and TCR in UKESM1.1 and UKESM1 from the 4xCO2 and 1%CO2 runs shows the long term-response to CO2 is similar between model configurations but is unable to test for a change in transient sensitivity to aerosol forcing. This would require dedicated historical aerosol-only simulations, which is planned in future work."*

**RC (copied again from above):** Similarly, the forcing values presented in Table 3 are not very convincing. The total anthropogenic forcing is indeed larger in UKESM1.1 (+1.84 W/m2) than UKESM1 (+1.76 W/m2). However, summing the components yields a smaller forcing for UKESM1.1 (+1.61 W/m2) than UKESM1 (+1.65 W/m2), and both of them are off by more than the difference between models. I don't think this data supports the conclusion that the change is aerosol forcing is key. Having comparable values for the period 1960-1990 would likely help.

**AR:** The component ERFs making up the total anthropogenic ERF in the fully coupled UKESM do not add up linearly as has been shown in a detailed assessment by O'Connor et al, 2021. This is due to the non-linearity in aerosol-cloud interactions but also aerosol-chemistry interactions (O'Connor et al 2021, O'Connor et al 2022). Changes to ozone precursors and CH4 have been shown to lead to an additional indirect aerosol forcing that would not be captured in an aerosol only ERF experiment set-up. An in-depth study of all the components of the total anthropogenic forcing is outside the scope

of this model documentation paper, we acknowledge however that this would be an interesting study to make in the future. In its absence for this manuscript, we have altered the text to as outlined above in our more general response to the reviewers Major Comment and have conducted additional aerosol ERF simulations which support the role of the less negative aerosol ERF in the more positive total anthropogenic forcing and the subsequent improved historical aerosol performance.

**Minor comments**

**RC:** Lines 122-127: paragraph requires clarification. If I understand correctly, r_c was set to 10 sm-1 in GC3.1, then mistakenly to 148.9 sm-1 in UKESM1 and then to 1 sm-1 in UKESM1.1. Clarify the motivation for using 1 sm-1 instead of 10 as in GC3.1? Insufficient SO2 dry deposition?

**AR:** The motivation comes from the literature already cited in the text (Garland,1978; Erisman et al. 1994; Zhang et al., 2003, Hardacre et al. 2021). SO2 is a highly soluble species and as such readily dissolves in water, supporting a low resistance value. Different ranges from 0.004 to 20 s m-1 appear in the literature, so 10 sm-1 as such isn't an incorrect value but lower values are also acceptable and further increase the SO2 dry deposition in the model. We have modified the text to make this clearer:

**L127:** *"Numerous studies (eg: Garland (1978), Erisman et al. (1994), Zhang et al. (2003)) indicate the resistance to SO2 deposition over the open ocean is minimal with reported resistance values ranging from 0.004 to 20.0 s m-1."*

**RC:** Lines 207-211: What is the impact on net TOA radiation?

**AR:** This is shown in the top panel of Figure 1 (P11) where the net TOA of the final tuned piControl is approximately 0.04Wm-2.

**RC:** Line 271: any reason for stopping at 462 years and not the recommended 500 years for DECK piControl?

**AR:** No. UKESM1.1 is well spun-up having been initialised after 111 years of the UKESM1 piControl plus running an additional 613 years in the model development cycle and another 70 years of the final frozen UKESM1.1 configuration. We ran the UKESM1.1 piControl long enough to cover our historical ensemble but felt it was unnecessary to run longer as these were not designed to be official CMIP simulations.

**RC:** Lines 275-276: "later period". Chosen because of the smaller drift or for other reason?

**AR:** The UKESM1.1 piControl was initialised from a later period in the UKESM1 piControl (see previous answer) than that used in Sellar et al. (2019). We use this later period in our analysis of the two models.

**RC:** Figure 3: HadCRUT5 reports SST over ice-free ocean, and surface air temperature over land and ice covered ocean. Was the same calculation done for the model output?

**AR:** We have used 'tas' or temperature at 1.5m everywhere in our calculation. Following the reviewers suggestion we have checked the impact of using just 'tas' versus a blended dataset of 'tas' and SST but find minimal difference (see below). We did discover in the process that in the original submission the 6 UKESM1 historical ensemble members used in Figure 3 were inconsistent with the 6 used elsewhere in the paper. This

has now been corrected in the revised manuscript and so while the UKESM1 values are slightly altered to previous it does not impact on the paper's findings or conclusions.

[Figure]

Figure: Surface air temperature anomalies using only 'tas'.

[Figure]

Figure: Surface air temperature anomalies using a blended 'tas' and SST dataset.

**RC:** Line 319: 1900 → 1901 for consistency with the figures. Similar on line 324.

**AR:** Now corrected (**L329-L350**)

**RC**: Lines 355-356: explain how globally averaged Nd and r_eff were calculated.

**AR:** These are 2d fields diagnosed at cloud-top output by the model. We have now clarified this in the text (see **L367/368 and L425, Figure 9 Caption** while the Caption of **Figure 6** now includes the following sentence: *"Both cloud droplet number concentration and effective radius represent cloud-top values"*

**RC:** Figure 6: how different are the starting values?

**AR:** We have now included an additional figure in the Supplementary Information (Figures S12) showing the historical timeseries of the absolute values of these aerosol and cloud variables. These figures show how the AOD is systematically higher in UKESM1.1 while the Nd is systematically lower. Differences for both variables are between 10-20%. We have referred to this figure **on L368** in the revised manuscript**.**

**RC:** Lines 365-372: are Nd anomalies really relevant if the clear-sky OSW anomalies are driving the surface temperature change?

**AR:** We contend that it is still relevant to show that the Nd response to historical forcing is smaller than in UKESM1 as it is the very different Nd climate in the PI that prevents a weaker ACI during the historical period, but it demonstrates that the model developments implemented in UKESM1.1 reduce the Nd response to anthropogenic forcings.

**RC:** Figure 9: explain how vertically averaged Nd was calculated.

**AR:** We have not plotted the vertically averaged Nd in this figure, it is Nd diagnosed at cloud-top. This is more comparable to the satellite derived quantity. We have made this now clear in the Figure 9 caption in the revised manuscript.

**RC:** Line 418: is the detrending actually needed? piControl looks stable in Figure 1b.

**AR:** Whether we can call the piControl "stable" depends on the quantity considered. For the ocean component, Figure 2 shows that both UKESM1.0 and UKESM1.1 display substantial centennial, internal variability. By applying detrending we have chosen to subtract most of this internal variability from the analysis of the historical simulations. Of course, other choices could have been made here. With our choice we ensure comparability with historical simulations from other models, as published in the literature.

**RC:** Line 425: "relatively large climate sensitivity" here, and "outside of the CMIP6 5-95% ranges" on line 552.

**AR:** This comment isn't very clear but we assume the reviewer is saying the 2 phrases are perhaps inconsistent. For clarity we have rephrased the former from "relatively large" to "high" **(L439).**

**RC:** Table 4: use a consistent number of decimals and verify that the net adds up, or explain why.

**AR:** Thank you for bringing this to our attention. We have now corrected the Table 4 to have a consistent number of decimal places. We also realised there was a typo in the "ACI cloud absorption" term for UKESM1, this should read -0.003 and not +0.003 and is now corrected. The components now add up to the net value.

**RC:** Line 602: "thesimulated" → "the simulated"
**AR:** Now corrected **(L616)**

**RC:** Lines 610-611: Table 3 currently doesn't support this assertion. See major comment above.

**AR:** We have now rephrased this sentence and paragraph, see response to Major Comment above.

**References:**

O'Connor, F. M., Abraham, N. L., Dalvi, M., Folberth, G. A., Griffiths, P. T., Hardacre, C., Johnson, B. T., Kahana, R., Keeble, J., Kim, B., Morgenstern, O., Mulcahy, J. P., Richardson, M., Robertson, E., Seo, J., Shim, S., Teixeira, J. C., Turnock, S. T., Williams, J., Wiltshire, A. J., Woodward, S., and Zeng, G.: Assessment of pre-industrial to present-day anthropogenic climate forcing in UKESM1, Atmos. Chem. Phys., 21, 1211–1243, https://doi.org/10.5194/acp-21-1211-2021, 2021.

O'Connor, F. M., Johnson, B. T., Jamil, O., Andrews, T., Mulcahy, J. P., & Manners, J. (2022). Apportionment of the pre-industrial to present-day climate forcing by methane using UKESM1: The role of the cloud radiative effect. *Journal of Advances in Modeling Earth Systems*, 14, e2022MS002991. https://doi.org/10.1029/2022MS002991

Hansen, J., et al. (2005), Efficacy of climate forcings, *J. Geophys. Res.*, 110, D18104, doi:10.1029/2005JD005776.

Shindell, D. Inhomogeneous forcing and transient climate sensitivity. *Nature Clim Change* **4**, 274–277 (2014). https://doi.org/10.1038/nclimate2136.

Salvi, P., Ceppi, P., & Gregory, J. M. (2022). Interpreting differences in radiative feedbacks from aerosols versus greenhouse gases. *Geophysical Research Letters*, 49, e2022GL097766. https://doi.org/10.1029/2022GL097766.

Sellar, A. A., Jones, C. G., Mulcahy, J. P., Tang, Y., Yool, A., Wiltshire, A., et al. (2019). UKESM1: Description and evaluation of the U.K. Earth System Model. *Journal of Advances in Modeling Earth Systems*, 11, 4513– 4558. https://doi.org/10.1029/2019MS001739